# AMP-activated protein kinase fortifies epithelial tight junctions during energetic stress via its effector GIV/Girdin

Nicolas Aznar[1], Arjun Patel[1], Cristina C Rohena[1], Ying Dunkel[1], Linda P Joosen[1], Vanessa Taupin[2], Irina Kufareva[3], Marilyn G Farquhar[2], Pradipta Ghosh[1,2,4]*

[1]Department of Medicine, University of California, San Diego, San Diego, United States; [2]Department of Cellular and Molecular Medicine, University of California, San Diego, San Diego, United States; [3]Skaggs School of Pharmacy and Pharmaceutical Sciences, University of California, San Diego, San Diego, United States; [4]Moores Cancer Center, University of California, San Diego, San Diego, United States

**Abstract** Loss of epithelial polarity impacts organ development and function; it is also oncogenic. AMPK, a key sensor of metabolic stress stabilizes cell-cell junctions and maintains epithelial polarity; its activation by Metformin protects the epithelial barrier against stress and suppresses tumorigenesis. How AMPK protects the epithelium remains unknown. Here, we identify GIV/Girdin as a novel effector of AMPK, whose phosphorylation at a single site is both necessary and sufficient for strengthening mammalian epithelial tight junctions and preserving cell polarity and barrier function in the face of energetic stress. Expression of an oncogenic mutant of GIV (cataloged in TCGA) that cannot be phosphorylated by AMPK increased anchorage-independent growth of tumor cells and helped these cells to evade the tumor-suppressive action of Metformin. This work defines a fundamental homeostatic mechanism by which the AMPK-GIV axis reinforces cell junctions against stress-induced collapse and also provides mechanistic insight into the tumor-suppressive action of Metformin.

*For correspondence: prghosh@ucsd.edu

Competing interests: The authors declare that no competing interests exist.

## Introduction

Epithelial cells normally display a polarized structure, with the membrane protein and organelle compositions differing between the basal and apical sides of the cell (*Kaplan et al., 2009*). This asymmetry between apical and basolateral compartments segregates structures, proteins, and organelle functions across polarized cells. Cell polarity is fundamental for both the architecture and function of epithelial tissues; its loss triggers organ dysfunction, neoplastic transformation and cancer progression, all via dysregulation of cell growth and division (*Martin-Belmonte and Perez-Moreno, 2012*).

Epithelial polarization requires the coordination of multiple fundamental cellular processes that are driven by their own set of unique signaling pathways and whose integration in space and time dictates overall epithelial morphogenesis (*St Johnston and Sanson, 2011*). Among the evolutionarily conserved pathways that control epithelial cell polarity, several collaborate to assemble, stabilize and turnover the cell-cell junctions, e.g. CDC42 and PAR proteins, such as the PAR3-PAR6-aPKC complex (*Wodarz and Nathke, 2007*), and pathways that regulate membrane exocytosis and lipid modifications (*St Johnston and Ahringer, 2010*; *Wodarz and Nathke, 2007*).

Besides the pathways mentioned above, regulation of polarity requires an additional signaling component which is triggered exclusively under conditions of energetic stress. Three studies (*Lee et al., 2007*; *Zhang et al., 2006*; *Zheng and Cantley, 2007*) published in 2006–2007

simultaneously reported a surprising role of AMP-activated protein kinase (AMPK) in the maintenance of epithelial cell polarity and barrier functions. Discovered in 1984, AMPK is unique in that it is a metabolic sensor protein which is activated during energetic stress and thereby, couples energy sensing to cell polarity by protecting cell junctions against stress-induced collapse. Using the polarized Madin Darby Canine Kidney Cell line (*MDCK*), it was demonstrated that AMPK is activated during calcium ($Ca^{2+}$)-induced tight junction (TJ) assembly (*Zhang et al., 2006*; *Zheng and Cantley, 2007*). Depletion of the AMPK catalytic α subunit or expression of a kinase-dead mutant of AMPK inhibits TJ assembly as indicated by a loss of transepithelial electrical resistance (TEER); the latter is a measure of paracellular ion flow which depends on TJ stability. Activation of AMPK with 5-aminoimidizole-4-carboxamide riboside (AICAR) partially protects TJs from disassembly induced by calcium depletion (*Zhang et al., 2006*; *Zheng and Cantley, 2007*). These findings closely followed another major revelation that the tumor suppressor LKB1 (Liver Kinase B1; also known as Serine/Threonine Kinase 11 – STK11) is a direct activator of AMPK (*Hawley et al., 2003*; *Hong et al., 2003*; *Shaw et al., 2004*; *Woods et al., 2003*), and that polarity defects precede the development of tumors in genetically modified mice with tissue-specific deletion of LKB1 (*Hezel et al., 2008*). Together, these discoveries established the first links between energetic stress, cell polarity and oncogenesis. Since then, multiple studies have reported the protective role of AMPK in maintaining cell-cell junctions across a variety of cell types in diverse tissue types [lung (*Garnett et al., 2013*), heart (*Castanares-Zapatero et al., 2013*), the blood-brain barrier (*Liu et al., 2014*; *Takata et al., 2013*), kidney (*Seo-Mayer et al., 2011*), intestine (*Spruss et al., 2012*)] while mounting a pathologic response to a variety of stressors, from bacterial invasion (*Patkee et al., 2016*) to ischemia (*Seo-Mayer et al., 2011*).

Although there is a wide consensus on the role of the LKB1-AMPK axis and in particular AMPK's role in reinforcing TJs and preserving cell polarity during adverse environmental changes, it remains largely unknown how this kinase actually accomplishes this. One study suggested that muscle myosin regulatory light chain (MRLC) may be the effector of AMPK during energetic stress in the fly (*Lee et al., 2007*), but those findings have since come into question (*Shackelford and Shaw, 2009*) because the phosphosites on MRLC do not conform to the optimal AMPK substrate motif found in all other established in vivo AMPK substrates. Furthermore, subsequent studies have demonstrated that the LKB1/AMPK pathway is *not required for* the maintenance of polarity during energetic stress in either flies (*Haack et al., 2013*; *Mirouse et al., 2013*) or fish (*van der Velden and Haramis, 2011*; *van der Velden et al., 2011*). Thus, despite the fact that it has been a decade since the first studies revealed AMPK's ability to preserve the epithelial architecture and function in the setting of energetic stress, effectors of AMPK that orchestrate these functions have not been identified. Here, we demonstrate that the multimodular polarity scaffold protein GIV (G-alpha interacting vesicle associated protein, a.k.a. Girdin) (see *Figure 1A*), is a novel substrate of AMPK, and define the molecular mechanisms by which the AMPK-GIV signaling axis protects the epithelium by stabilizing TJs and preserving cell polarity when challenged with energetic stress. Findings also reveal how deregulation of this pathway fuels the growth of tumor cells under energetic stress.

## Results and discussion

### AMPK binds and phosphorylates GIV at residue S245

GIV regulates epithelial cell polarity and morphogenesis (*Bhandari et al., 2015*; *Houssin et al., 2015*; *Sasaki et al., 2015*); it's role at cell-cell junctions has been attributed to its ability to bind PAR3 (*Sasaki et al., 2015*) and the cadherin-catenin complexes (*Houssin et al., 2015*), and accelerate nucleotide exchange on (i.e. activate) the trimeric G-protein α subunit, Gαi via its C-terminal GEF motif; *Figure 1A* (*Sasaki et al., 2015*). An examination of GIV's sequence revealed the presence of an evolutionarily conserved optimal substrate recognition site for AMPK [FxR/KxxS/TxxxL (*Banko et al., 2011*; *Hardie et al., 2016*; *Marin et al., 2015*)] within the N-terminus of GIV (aa 239–250; Accession no. Q3V6T2) (*Figure 1B–C*). We asked if AMPK recognizes (binds and phosphorylates) GIV as a substrate and phosphorylates Ser 245 (S245) within the consensus. To investigate if AMPK binds GIV, we used two complementary approaches. First, co-immunoprecipitation assays using Cos7 cells expressing myc-tagged AMPKα2 confirmed that GIV co-immunoprecipitates with AMPK (*Figure 1D*). Second, GST-pulldown assays confirmed that the N-terminal 440 aa of GIV

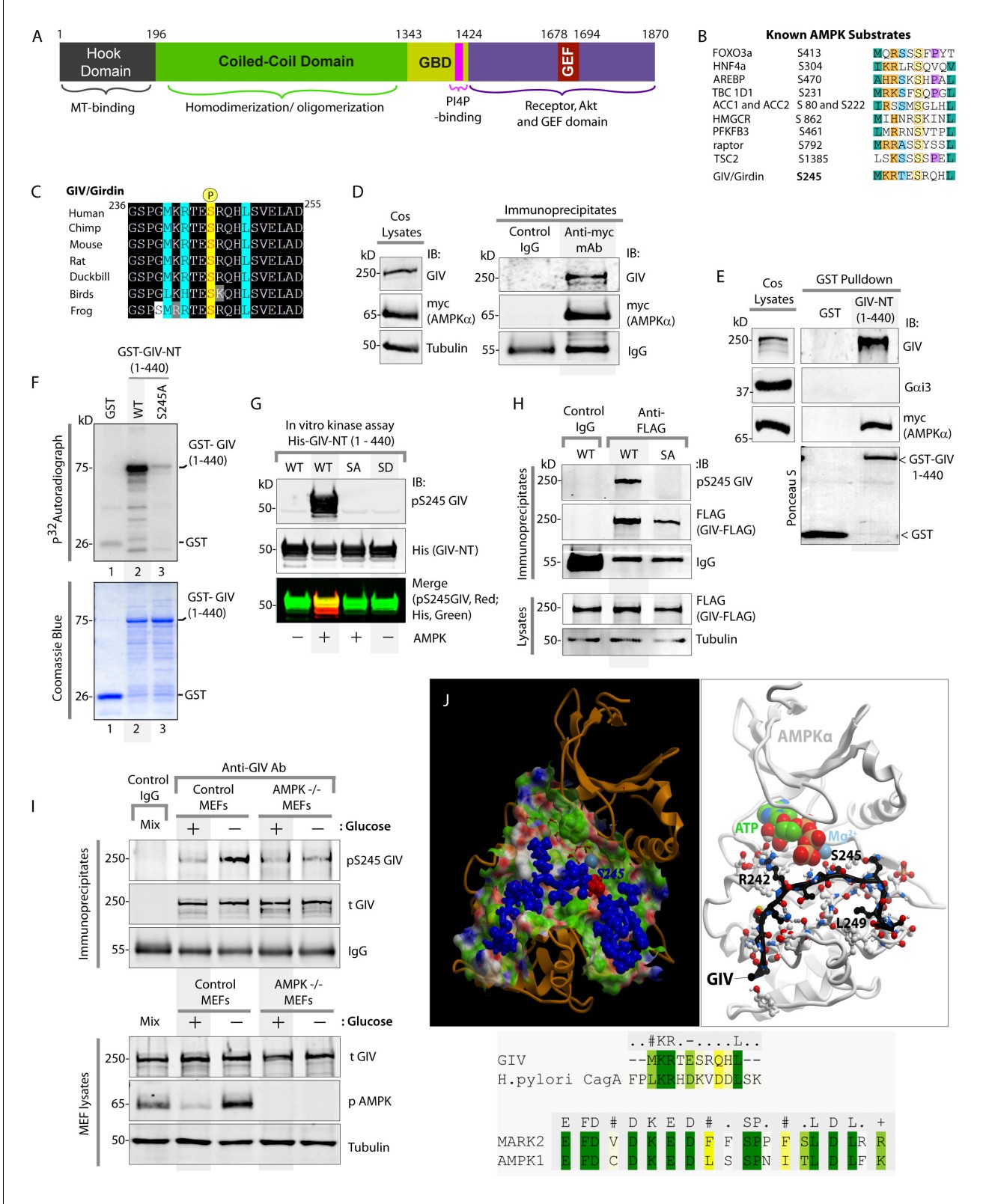

**Figure 1.** AMPK binds and phosphorylates GIV at Ser (**S**) 245. (**A**) Schematic showing the functional modules of the multimodular signal transducer GIV. From the N- to the C-terminus the domains are– a Hook-domain (grey) which binds microtubules (*Simpson et al., 2005*); a long coiled-coil domain (green) assists in homo/oligomerization (*Enomoto et al., 2005*); a Gα-binding domain (GBD; yellow) which constitutively binds Gαi/s proteins (*Le-Niculescu et al., 2005*); a PI(4)P-binding motif (pink) which enables GIV to bind PI4P-enriched membranes at the Golgi and the PM (*Enomoto et al.,*

*Figure 1 continued*

*2005*); an evolutionarily conserved GEF motif (red) which binds and activates Gαi (*Garcia-Marcos et al., 2009*) and inactivates Gαs (*Gupta et al., 2016*), and releases 'free' Gβγ from both. The C-terminal ~200 aa of GIV (purple) also has key domains that enable GIV to bind and remodel actin (*Enomoto et al., 2005*), bind and enhance phosphorylation of Akt (*Anai et al., 2005*; *Enomoto et al., 2005*), bind ligand-activated RTKs (*Ghosh et al., 2010*; *Lin et al., 2014*), and bind and activate Class 1 PI3-Kinases (*Lin et al., 2011*). (B) Consensus phosphorylation site for previously identified substrates of AMPK are aligned with the putative AMPK substrate site in human GIV. Conserved residues are highlighted with colors. (C) The sequence encompassing the putative AMPK substrate motif was aligned among various species using ClustalW. Conserved residues are shaded in black and similar residues in gray. The consensus residues within the sequence are highlighted in blue. The residue, Ser(S)245 which was predicted to be phosphorylated by AMPK is highlighted in yellow. (D) Immunoprecipitations were carried out on lysates of Cos7 cells expressing myc-AMPKα2 using anti-myc mAb. Immune complexes were analyzed for endogenous GIV and myc (AMPKα2) by immunoblotting (IB). (E) Lysates of Cos7 cells expressing myc-AMPKα2 were used as a source of AMPK in pulldown assays with bacterially expressed GST or GST-GIV-NT (aa 1–440; which includes S245) immobilized on glutathione beads. Bound proteins were analyzed for myc (AMPKα), Gαi3 (negative control; because this G protein binds GIV's C-terminus, not N-terminus) and endogenous GIV (positive control; because GIV homo-oligomerizes via its NT) by immunoblotting (IB). (F) In vitro kinase assays were carried out using recombinant AMPK heterotrimers (α2/β/γ) and bacterially expressed and purified GST-GIV-NT (1–440) proteins or GST alone (negative control) and γ -32P [ATP]. Phosphoproteins were analyzed by SDS-PAGE followed by autoradiography (top). Equal loading of substrate proteins was confirmed by staining the gel with Coomassie blue (bottom). AMPK phosphorylated GST-GIV-NT WT, but not the non-phosphorylatable SA mutant or GST alone. (G) Biochemical validation of a phosphospecific rabbit polyclonal antibody which detects GIV exclusively when it is phosphorylated at S245. In vitro kinase assays were carried out as described above and incubated in the presence of cold ATP. Phosphoproteins were analyzed for pS245-GIV and His (GIV-NT) by immunoblotting (IB). (H) In cellulo kinase assays were carried out in Cos7 cells co-expressing GIV-FLAG (WT or SA mutant) and myc-AMPKα constructs, and stimulating AMPK by glucose deprivation for 6 hr prior to lysis. GIV was immunoprecipitated from these lysates using anti-FLAG mAb and analyzed for phosphorylation of GIV at S245 by immunoblotting (IB) with anti-pS245-GIV and FLAG (GIV-Flag). GIV-WT, but not GIV-SA is phosphorylated at S245 in cells responding to energetic stress. (I) AMPK-/- and control MEFs were subjected or not to energetic stress by exposing them to growth conditions with (+) or without (-) glucose for 4 hr prior to lysis. GIV immunoprecipitated from equal aliquots of lysates (lower panel) were analyzed for total (t) and phosphorylated (pS245) GIV by immunoblotting. Representative blots are shown (n = 3). (J) *Top*: Homology model of GIV-bound AMPKα generated using the solved crystal structure of constitutively active Par1-MARK2 (a member of the AMPK family of kinases) in complex with the CagA protein encoded by pathologic strains of *Helicobacter pylori* [PDB: 3IEC (*Nesic et al., 2010*)] as template. *Bottom*: The target:template alignment is shown for the GIV peptide (with *H. pylori* CagA protein) and AMPK (with MAPK2, partial alignment of binding site residues).

[which contains the residue S245] is sufficient for binding AMPKα2 (*Figure 1E*). These results demonstrate that GIV binds AMPKα both in vitro and in cells and that the interaction is mediated via GIV's N-terminus.

To investigate if AMPK phosphorylates GIV, we carried out in vitro and in cellulo kinase assays and generated an affinity-purified rabbit polyclonal phosphoS245-GIV antibody to specifically detect the phosphoprotein by immunoblotting. Autoradiographs of in vitro kinase assays carried out using γP-*32 ATP*, recombinant AMPKα2 heterotrimers and bacterially expressed N-terminal fragment (aa 1–440) of GIV showed that AMPK phosphorylates GST-GIV-NT WT (1–440 aa) but not a non-phosphorylatable mutant protein in which the Ser at 245 is mutated to Ala (A) (S245A; henceforth referred to as SA) (*Figure 1F*). Identical results were obtained when in vitro kinase assays were carried out as above, except by replacing γP-*32 ATP with* non-radioactive ATP and analyzed by immunoblotting with the anti-pS245-GIV antibody (*Figure 1G*). The phospho-specific antibody also showed a high degree of specificity; it did not detect non-phosphorylated GIV-WT or the phospho-mimicking mutant in which S245 is mutated to Asp(D) (S245D; henceforth referred to as SD) (*Figure 1G*). To confirm if AMPK phosphorylates GIV at S245 in cells, we carried out in cellulo kinase assays in cells coexpressing both GIV-FLAG (substrate) and myc-AMPKα (kinase). Because AMPK is activated by increasing concentrations of AMP during metabolic stress induced by glucose starvation (*Hardie, 2004*; *Tzatsos and Tsichlis, 2007*), we triggered activation of AMPK by growing the cells in glucose-free medium prior to lysis and by immunoprecipitating GIV and analyzing it for phosphorylation at S245 by immunoblotting (*Figure 1H*). Phosphorylation of GIV at S245 was detected exclusively in the glucose-starved cells expressing GIV-WT, but not in those expressing GIV-SA (*Figure 1H*). That AMPK is required for phosphorylation of GIV at S245 after energetic stress was further confirmed using mouse embryonic fibroblasts (MEFs); such phosphorylation was induced by glucose deprivation in control MEFs, but not in MEFs derived from AMPK-/- mice (*Figure 1I*). These results demonstrate that AMPK phosphorylates GIV at S245 in vitro and in cells. Furthermore, a homology model of GIV's N-terminal sequence bound to the catalytic α-subunit of AMPK generated using a previously solved crystal structure of CagA in complex with the AMPK-related kinase,

MARK2 [PDB: 3IEC (*Nesic et al., 2010*)], showed that the residues within and flanking S245 and the consensus substrate recognition sequence are compatible with AMPK's ability to bind and phosphorylate GIV at Ser 245 (*Figure 1J*).

These findings are in keeping with results from the first large-scale high-throughput (HTP) Mass Spectrometry analysis (LC-ESI-MS/MS) aimed at mapping human protein-protein interactions (*Ewing et al., 2007*) in which GIV co-immunoprecipitated with FLAG-tagged AMPK in HEK293 cells (*Ewing et al., 2007*) and two prior HTP MS studies aimed at mapping the human phosphoproteome (*Schweppe et al., 2013*; *Zhou et al., 2013*) which also reported that S245 on endogenous GIV is phosphorylated in cells. We conclude that GIV is a bona fide substrate of AMPK and that S245 is a specific substrate site for the kinase.

## GIV phosphorylated at S245 localizes to cell-cell junctions

Next we asked where GIV, and more specifically GIV that is phosphorylated at S245 (henceforth referred to as pS245-GIV), localizes in cells responding to energetic stress. We studied this by carrying out immunofluorescence studies on monolayers of Type II MDCK cells which are a widely used cell-based model system to study epithelial cell polarity, junctional integrity and epithelial morphogenesis (*Dukes et al., 2011*). Type II MDCK cells have been used as the primary model system in the initial studies that revealed the role of AMPK in the stabilization of TJs (*Zhang et al., 2011*; *Zhang et al., 2006*; *Zheng and Cantley, 2007*). While GIV was not detected at cell-cell contact sites in fully polarized domed MDCK monolayers at steady-state under basal (normoglycemic) conditions, it localized transiently to cell-cell junctions during energetic stress (*Figure 2A*). Active AMPK also localized transiently to cell-cell contact sites exclusively after energetic stress at specialized zones of the TJs, called tricellular TJs (*Figure 2—figure supplement 1*). pS245-GIV was not detected at the cell-cell contact sites either in isolated single cells or in fully polarized domed MDCK monolayers under normal glucose conditions, but localized to the cell-cell junctions transiently when single cells assembled junctions during the course of their growth into confluent monolayers (*Figure 2B*). Although pS245-GIV was not detected at the junctions of fully polarized domed monolayers, it was prominently detected once the monolayers were subjected to energetic stress with glucose starvation (*Figure 2B*). pS245-GIV was also readily detected at cell-cell junctions when domed monolayers were exposed to low-calcium states with the $Ca^{2+}$-chelator EGTA, indicating that such localization is $Ca^{2+}$-independent. Both conditions, i.e., low-calcium states (EGTA-induced) and energetic stress states (induced by glucose deprivation) were accompanied by activation of AMPK (*Figure 2—figure supplement 2*). Using cross-sections of Z-stacked images of MDCK cells treated with EGTA, we confirmed that pS245-GIV colocalizes preferentially with Occludin, a marker of tight junctions (TJs), and only occasionally with β-Catenin, a marker of adherens junctions (AJs) (*Figure 3*). These results demonstrate that pS245-GIV preferentially localizes to the TJs and that such localization is seen exclusively during TJ turnover; localization is seen both during TJ assembly as cells come in contact to form a monolayer and during TJ-disassembly as monolayers collapse in response to energetic stress or $Ca^{2+}$-depletion. These findings raise the possibility that pS245-GIV may serve as one of the effectors of AMPK at the TJs which enables the kinase to enhance junctional stability exclusively during stress-induced junction turnover. That little or no pS245-GIV was observed at cell-cell contact sites in fully polarized domed monolayers at steady-state indicates one of the two possibilities: (i) the AMPK-GIV axis, of which pS245-GIV is a product, may not have a role at the TJs under physiologic conditions, but is required to fortify TJs exclusively under conditions of energetic stress, or (ii) pS245-GIV may be required for the maintenance of TJs, but cannot be detected because of rapid dephosphorylation and a short half-life at the junctions. Finally, that GIV-SA expressing cells have normal TEER at steady-state (no energetic stress, normal calcium, normal serum; *Figure 4C*), but fail to 'recover' TEER when they are returned to NCM growth conditions after an initial exposure to LCM (*Figure 4D*) suggests that in the absence of the AMPK-GIV axis energetic stress and disassembly of junctions triggered by low calcium may set in motion some yet unknown pathways in an irreversible manner, perhaps culminating in detachment and/or apoptosis.

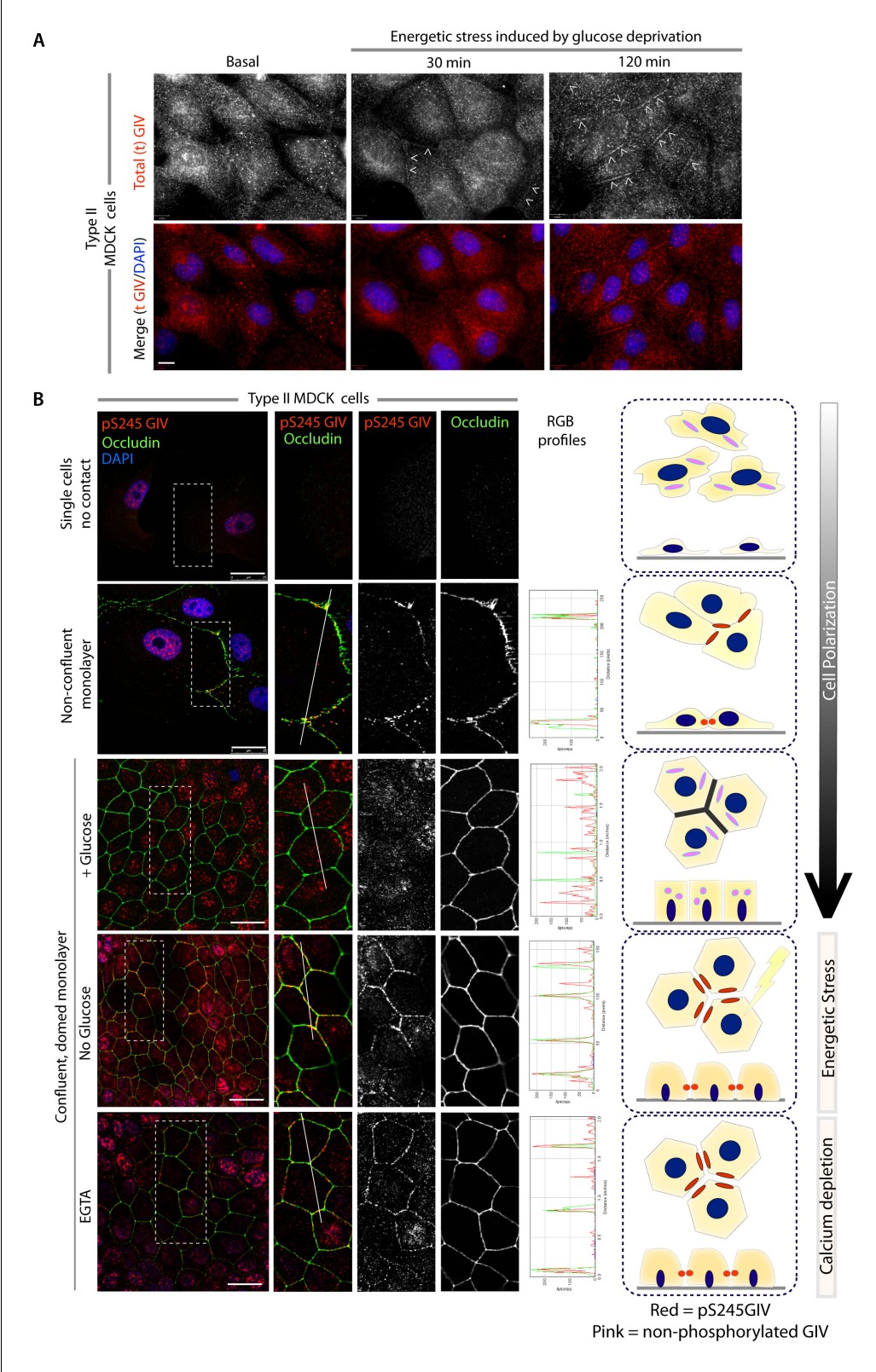

**Figure 2.** GIV and S245-phosphorylated GIV localize to cell-cell junctions. (**A**) MDCK cells were grown on glass cover slips, exposed or not to energetic stress (glucose deprivation for 30 and 120 min), and subsequently fixed and stained for total (t) GIV (red) and DAPI (blue; nuclei) and analyzed by confocal microscopy. Representative confocal images are shown. In domed monolayers at basal condition, GIV is absent from cell-cell junctions. However, GIV is detected at cell-cell junctions (arrowheads) after energetic stress induced by glucose deprivation. Scale bar = 10 μm. (**B**) MDCK cells

*Figure 2 continued on next page*

*Figure 2 continued*

were grown on glass cover slips, at various stages during their growth phase: from single-cells (top), to non-confluent monolayers (middle), to confluent domed monolayers (bottom). They were fixed and stained for Occludin (a TJ marker; green), pS245-GIV (red) and DAPI (blue; nuclei) and analyzed by confocal microscopy. Confluent monolayers were either maintained in complete medium in the presence of glucose (+ Glucose), or subjected to energetic stress by glucose deprivation (No Glucose) or calcium deprivation (EGTA). Representative fields are shown on the left. RGB plots, generated using ImageJ on the right assess the degree of colocalization between pS245-GIV and Occludin along the lines in the corresponding images during cell polarization are shown in the middle. Schematics summarizing the staining pattern in each condition are shown on the right. Scale bar = 25 μm.

The following figure supplements are available for figure 2:

**Figure supplement 1.** Active AMPK localizes to specialized regions of the TJs called tricellular TJs exclusively after energetic stress.

**Figure supplement 2.** AMPK is activated when type II MDCK cells are exposed to energetic stress or low-calcium growth conditions.

## Phosphorylation of GIV at S245 is essential for junctional integrity and epithelial morphogenesis

To determine the role of pS245-GIV at TJs in MDCK cells, we first compared the levels of expression of endogenous GIV by immunoblotting lysates of Type I and II MDCK cells which present different physiological properties such as junction integrity and stability (*Barker and Simmons, 1981*). Expression of GIV was significantly reduced (~by 80–85%) in Type II MDCK cells, which present 'leakier' junctions compared to Type I MDCK cells (*Rothen-Rutishauser et al., 1998*) (*Figure 4—figure supplement 1*). Such low levels of GIV expression have previously been reported by others [GIV in MDCK II cells could be detected by immunoblotting exclusively after enrichment by immunoprecipitation (*Sasaki et al., 2015*)]. Despite multiple attempts at depleting the residual low amounts of GIV by shRNA (lentiviral vectors), we failed to generate cell lines stably and reliably depleted of GIV, suggesting that the already low levels of endogenous GIV may be essential for the survival of MDCK cells in monolayer cultures. As an alternative approach, to study how phosphorylation of GIV at S245 affects cell polarity we generated MDCK II cell lines stably expressing FLAG-tagged full length GIV constructs at levels ~ 2 to 3 -fold above the low levels of the endogenous protein (*Figure 4A*): GIV-WT (which models the physiologic state allowing cycles of phospho-dephosphorylation at S245); the non-phosphorylatable SA (which models the constitutively dephosphorylated state); the phosphomimicking SD mutants (which models the constitutively phosphorylated state at S245). The functional integrity of TJs, as determined by measuring paracellular permeability of ions assessed by transepithelial electrical resistance (TEER) across fully polarized domed monolayers (*Figure 4B*) were similar between the parental and various Type II MDCK-GIV stable cell lines at steady-state (normal glucose and calcium); as expected, all of them were significantly 'leakier' than Type I MDCK cells (*Figure 4C*). However, when these cells were exposed to low-$Ca^{2+}$ media (*Figure 4D*) or deprived of glucose (*Figure 4E*) the drop in TEER when was significantly more across MDCK-GIV-SA monolayers and significantly less across MDCK-GIV-SD monolayers compared to MDCK-GIV-WT. When normocalcemic conditions were restored, MDCK-GIV-WT and SD cells regained their baseline TEER, whereas MDCK-GIV-SA cells did not (*Figure 4D*). The MDCK-GIV-SD monolayers also showed a smaller drop in TEER (25%, compared to 50% in MDCK-GIV-WT monolayers) when exposed to EGTA (*Figure 4—figure supplement 2A*), whereas the MDCK-GIV-SA monolayers failed to regain their TEER as efficiently as MDCK-GIV-WT cells after a $Ca^{2+}$ switch (*Figure 4—figure supplement 2B*). These findings indicate that although all MDCK-GIV cell lines make stable TJs in normal growth conditions, the MDCK-GIV-SA monolayers are more vulnerable and the MDCK-GIV-SD monolayers are more resilient to $Ca^{2+}$-depletion or energetic stress compared to GIV-WT cells.

The patterns of vulnerability versus resilience between the MDCK-GIV-SA and SD cell lines were also observed when the morphological integrity of TJs was assessed by immunofluorescence. Under normoglycemic and normocalcemic conditions, MDCK-GIV cell lines were indistinguishable from the parental Type II cells and from each other. They all grew into confluent monolayers of fully polarized cells with intact TJs and AJs, as determined by immunofluorescence studies looking at the localization of Occludin (*Figure 4—figure supplement 3*), E-Cadherin, ZO-1 or β-Catenin (data not shown). However, clear differences emerged when the localization of TJ and AJ proteins were analyzed in these cells after exposing them to conditions that trigger disassembly of cell-cell junctions by either

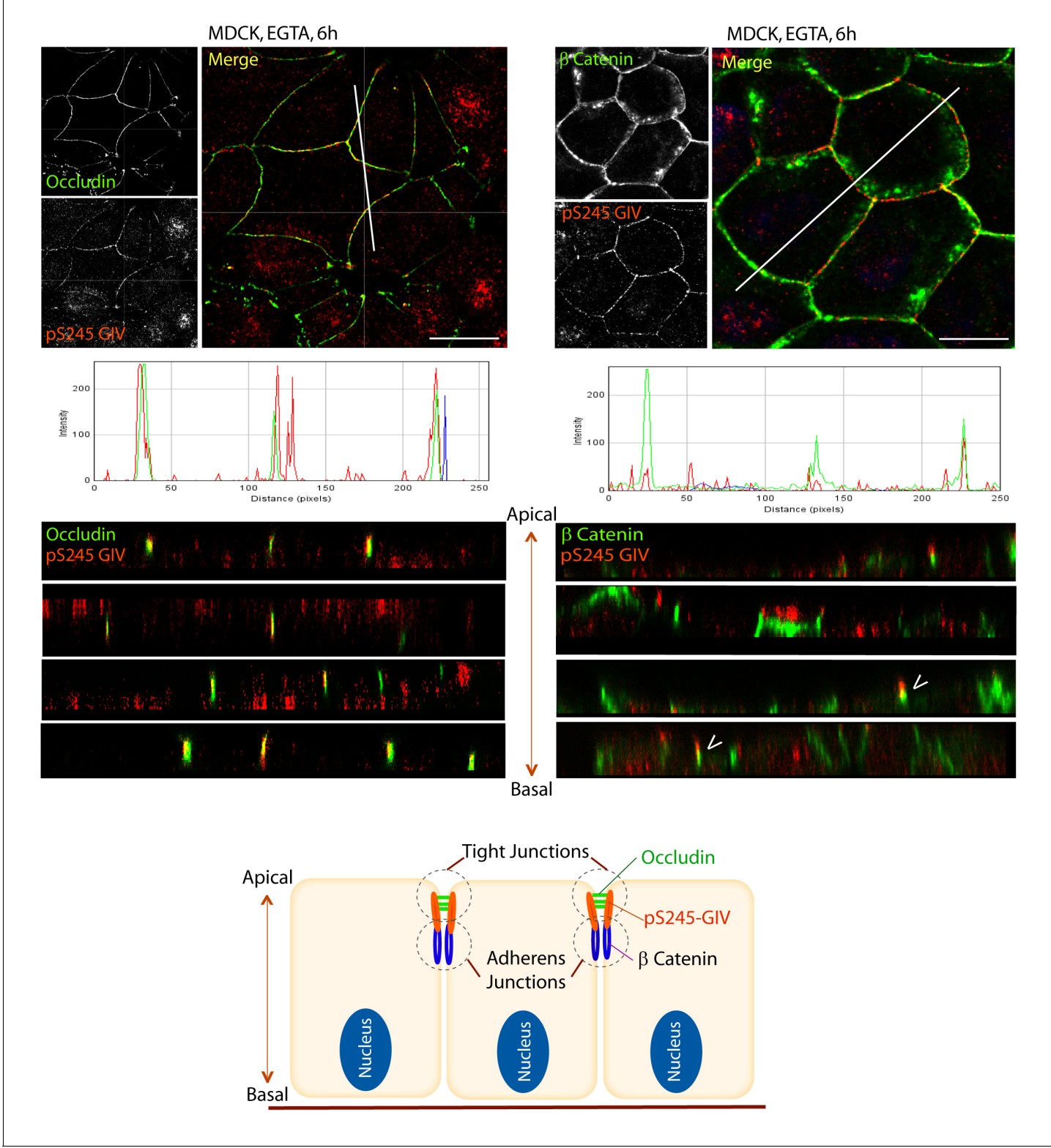

**Figure 3.** S245-phosphorylated GIV (pS245-GIV) localizes preferentially to tight junctions (TJs). MDCK cells were grown to full confluency into domed monolayers prior to exposing them to the Ca²⁺ chelator EGTA for 6 hr prior to fixation. Fixed cells were co-stained for pS245-GIV (red) and either the TJ-marker Occludin (Left; green) or the AJ-marker β-Catenin (Right; green) and analyzed by confocal microscopy (z-stack projections and x-z cross-sections). *Top*: Representative confocal images are shown, each taken at the level of the TJs (marked by Occludin; left) and AJs (marked by β-Catenin; right). *Middle*: RGB profiles showed that pS245 GIV colocalized with Occludin but not β-Catenin. *Bottom*: Sections through the 3D reconstruction of a
*Figure 3 continued on next page*

*Figure 3 continued*

confocal Z-stack confirms that, in most instances the pS245-GIV colocalizes with the TJ marker Occludin, and lies just apical to the AJ marker β-Catenin. In some cases (arrowheads), the lower pole of the pS245-GIV signal partially colocalizes with the upper pole of the β-Catenin signal. Schematic summarizes the localization of pS245-GIV and its relationship to the TJ and AJs. Scale bar = 25 µm.

depriving them of $Ca^{2+}$ (with EGTA; *Figure 4F*, *Figure 4—figure supplement 4*) or by subjecting them to energetic stress (by glucose starvation; *Figure 4—figure supplement 5*). Both conditions revealed the early disruption of TJs and AJs and increased vulnerability of MDCK-GIV-SA cells compared to control MDCK parental cells or the MDCK-GIV-WT cells. These cells exhibited increased actin remodeling and formed stress fibers compared to control or GIV-WT cells, indicative of junctional instability, loss of polarity and cell-flattening (*Figure 4—figure supplement 5*). By contrast, the MDCK-GIV-SD cells were more resilient, in that junctional markers localized to cell-cell junctions despite prolonged exposure to conditions that disrupt junctions (*Figure 4F*; *Figure 4—figure supplements 4–5*). These findings indicate that an interruption in the AMPK-GIV axis during energetic stress, as modeled in the MDCK-GIV-SA cells in which AMPK cannot phosphorylate S245 on GIV, made junctions more vulnerable, leakier, and triggered loss of cell polarity. Under the same set of conditions hyperactivation of the AMPK-GIV axis, as modeled in the MDCK-GIV-SD cells which mimics constitutive phosphorylation at S245 on GIV, resisted stress-induced loss of junctional integrity and preserved barrier functions and cell polarity in the face of stress. We conclude that phosphorylation of GIV at S245 by AMPK is largely dispensable under normal growth conditions but essential for TJ integrity exclusively during energetic stress. This single phosphoevent not only resists stress-induced junctional collapse, but also favors their reassembly during recovery.

Because destabilization of the epithelial barrier alters epithelial morphogenesis (*Pollack et al., 1998*; *Roignot et al., 2013*), next we asked if the contrasting junctional properties of MDCK-GIV monolayers affect cell behavior during cystogenesis within *3D Matrigel* cultures; the latter is a stringent assay for testing the integrity of polarity pathways. When cultivated in 3D collagen matrix, which limits the availability of glucose, nutrients and oxygen by diffusion only, MDCK cells with intact polarity pathways form cysts comprised of polarized cells lining a single central lumen (*O'Brien et al., 2002*). When one or more of the polarity pathways are disrupted, cysts display abnormal features, e.g., deformed or multiple lumens, elongated structures, tubules or other abnormal shapes or sizes. MDCK-parental and MDCK-GIV-WT cells morphed into cysts with a single central lumen and infrequently into small tubular structures (*Figure 4G–J*). MDCK-GIV-SD cells formed cysts that were consistently smaller in diameter (by ~30–40%) with a smooth edged central lumen (*Figure 4I*), but with a notable absence of tubular structures. By contrast, MDCK-GIV-SA cells morphed into larger cysts invariably featuring multiple lumens and abundant complex branched tubular structures (*Figure 4G,J*). These data indicate that phosphorylation on GIV S245 is a key determinant of normal epithelial morphogenesis– phosphorylation favors polarized normal cysts, whereas absence of phosphorylation favors branching tubules and multi-lumen structures that are associated with loss of cell polarity.

## Stabilization of tight junctions by AMPK requires phosphorylation of GIV at S245

Previous studies (*Zhang et al., 2006*; *Zheng and Cantley, 2007*) have shown that inhibition of AMPK, either by genetic manipulation or by chemicals like Compound C makes TJs vulnerable to disruption, whereas pharmacologic activation of AMPK with AICAR or Metformin stabilizes TJs. We used these modulators of AMPK on the MDCK-GIV stable cell lines to investigate if the previously observed role of AMPK in the protection of epithelial TJs is mediated via its ability to phosphorylate GIV at S245. If such phosphorylation is essential, we expected that pre-treatment of monolayers with AMPK agonists AICAR or Metformin would not protect TJs formed by MDCK-GIV-SA cells no matter how efficiently AMPK is activated. Conversely, we expected that treatment of monolayers with the AMPK inhibitor Compound C would not disrupt TJs assembled by MDCK-GIV-SD cells, in which the need for AMPK has been bypassed by expression of a mutant that mimics constitutive activation of the pathway. We found such is indeed the case because compared to MDCK-GIV-WT cells, the loss of functional integrity of TJs (as determined by a drop in TEER) after short (2 hr) exposures to

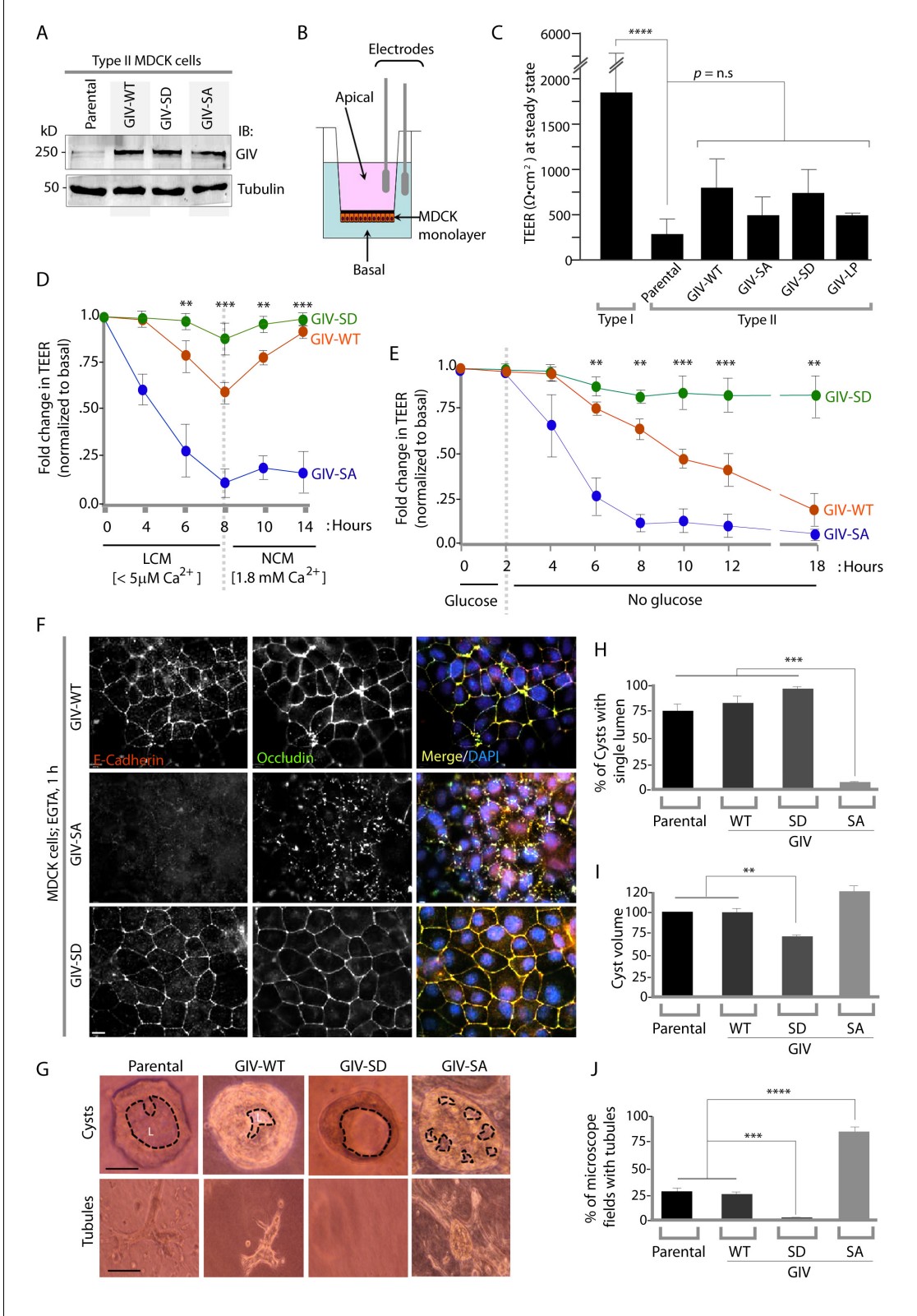

**Figure 4.** Phosphorylation of GIV at S245 stabilizes tight junctions (TJs) under low-calcium states and is essential for epithelial barrier function and morphogenesis. (**A**) Whole cell lysates of MDCK cells stably expressing wild-type GIV (GIV-WT), or non-phosphorylatable (S→A; GIV-SA), or phospho-mimicking (S→D; GIV-SD) mutants of GIV were analyzed for GIV and tubulin by immunoblotting. (**B–C**) Transepithelial electrical resistance (TEER) was measured using a Millicel-ERS resistance meter across fully polarized domed monolayers of various MDCK cell lines grown to domed confluency in

*Figure 4 continued on next page*

*Figure 4 continued*

Transwell inserts (schematic shown in **B**) in the presence of full growth media with normal calcium. Bar graphs (**C**) display the TEER measured across each cell line. As expected, type I MDCKs exhibit higher TEER than type II cells. No significant differences were noted between the MDCK-GIV cell lines stably expressing the various GIV constructs. n = 3. Results are expressed as ± SEM. ****p<0.0001. (**D–E**) Changes in TEER was measured across monolayers of various MDCK cell lines during exposure to low-calcium media (LCM; left), followed by switching to normal calcium media (NCM; right) or during energetic stress when exposed to media without glucose. Graphs in **D** and **E** show that compared to MDCK-GIV-WT cells, TEER rapidly dropped across MDCK-GIV-SA monolayers exposed to LCM (**D**) as well as energetic stress (**E**) but relatively preserved in MDCK-GIV-SD cells. This drop is rapidly and completely reversed upon exposure to NCM in MDCK-GIV-WT and SD cells, but remains impaired in MDCK-GIV-SA cells. n = 3. Results are expressed as ± SEM. **p<0.01; ***p<0.001; **p<0.01. (**F**) MDCK-GIV cell lines were grown to full confluency into domed monolayers, treated with EGTA for 1 hr, and subsequently fixed. Fixed cells were stained for E-cadherin, Occludin, and DAPI (nuclei; blue), and analyzed by confocal microscopy. Images displayed are representative of 13–15 HPF images captured at 60X mag in each cell line. Preservation of TJs and AJs, as visualized using Occludin and E-Cadherin as markers, was significantly higher in MDCK-GIV-WT and MDCK-GIV-SD [65–90% of the imaged surface area; n = 15 randomly imaged fields in each cell line], but not in MDCK-GIV-SA cell lines [0–4% of the imaged surface area; n = 13 randomly imaged fields; p<0.001]. Scale bar = 10 μm. See also *Figure 4—figure supplements 2 and 3* for the findings at baseline and at 4 hr time point. (**G**) Parental MDCK cells and various MDCK-GIV cell lines were seeded and grown in collagen-containing matrix for 2 weeks and analyzed for the formation of cyst and tubular structures by light microscopy. Representative cysts and tubular structures are shown for each cell line. L = lumen. Scale bar = 50 μm. (**H–J**) Bar graphs display the % of cysts with single lumens (Y axis; **H**), cyst volume (Y axis; **I**) and % fields with tubule formations (Y axis; **J**) seen in each cell line in G. Absolute numbers for cyst volume were normalized to parental cells (set to 100%). Three independent experiments comprising 450–600 cysts per cell line are summarized. Results are expressed as ± SEM. **p<0.01; ***p<0.001; ****p<0.0001.

The following figure supplements are available for figure 4:

**Figure supplement 1.** A comparison of levels of GIV expression in Type I vs Type II MDCK cells.
**Figure supplement 2.** Phosphorylation of GIV at S245 is required for maintenance of TEER in the absence of calcium in growth media, as well as for recovery of TEER after calcium switch.
**Figure supplement 3.** A comparison of tight junction morphology in parental and the various Type II MDCK-GIV cell lines.
**Figure supplement 4.** Phosphorylation of GIV at S245 stabilizes tight junctions (TJs) under low-calcium states.
**Figure supplement 5.** Phosphorylation of GIV at S245 stabilizes tight junctions (TJs) in cells exposed to energetic stress.

Compound C was significantly less in MDCK-GIV-SD cells and greater in MDCK-GIV-SA cells (*Figure 5A*). As anticipated, the protective actions of Metformin (*Figure 5B*) or AICAR (*Figure 5C*) on TJs of cells exposed to low $Ca^{2+}$ media was seen in MDCK-GIV-WT, but not in MDCK-GIV-SA cells. Immunofluorescence studies confirmed that the increased susceptibility and resilience of GIV-SA and GIV-SD cells, respectively, to the AMPK inhibitor Compound C correlate with the early and late loss of structural integrity of TJs and AJs, as determined by the loss of localization of junctional proteins from cell-cell contact sites (*Figure 5D*). Prolonged exposure of monlayers to Compound C showed that MDCK-GIV-SD cells continue to stay attached as monolayers, long after MDCK-GIV-WT and SA cell lines have detached and floated in clumps of dead cells, indicating that the GIV-SD cell line is relatively insensitive to the AMPK-inhibitor Compound C (*Figure 5—figure supplement 1*). The findings in the MDCK-GIV-SA cells indicate that phosphorylation of GIV at S245 is required for the protective actions of AMPK activators on TJ integrity. The findings in the MDCK-GIV-SD cells indicate that phosphorylation of GIV at S245 is sufficient for offering protection against and escaping the disruptive actions of AMPK inhibitors on TJs (see schematic and legend; *Figure 5E*). Because Compound C, the only available cell permeable effective inhibitor of AMPK can also inhibit other kinases besides AMPK (*Bain et al., 2007*; *Vogt et al., 2011*), there is a possibility that GIV may interplay with kinases other than AMPK to maintain TJ integrity. We conclude that pS245-GIV, which is generated only when the AMPK-GIV axis is intact, is both necessary and sufficient to fortify TJs, avoid junctional collapse and preserve cell polarity in the face of energetic stress. We further conclude that a significant part of the junction-stabilizing effects of AMPK agonists AICAR and Metformin during energetic stress (*Zhang et al., 2006*; *Zheng and Cantley, 2007*) are mediated by AMPK via its downstream effector, pS245-GIV.

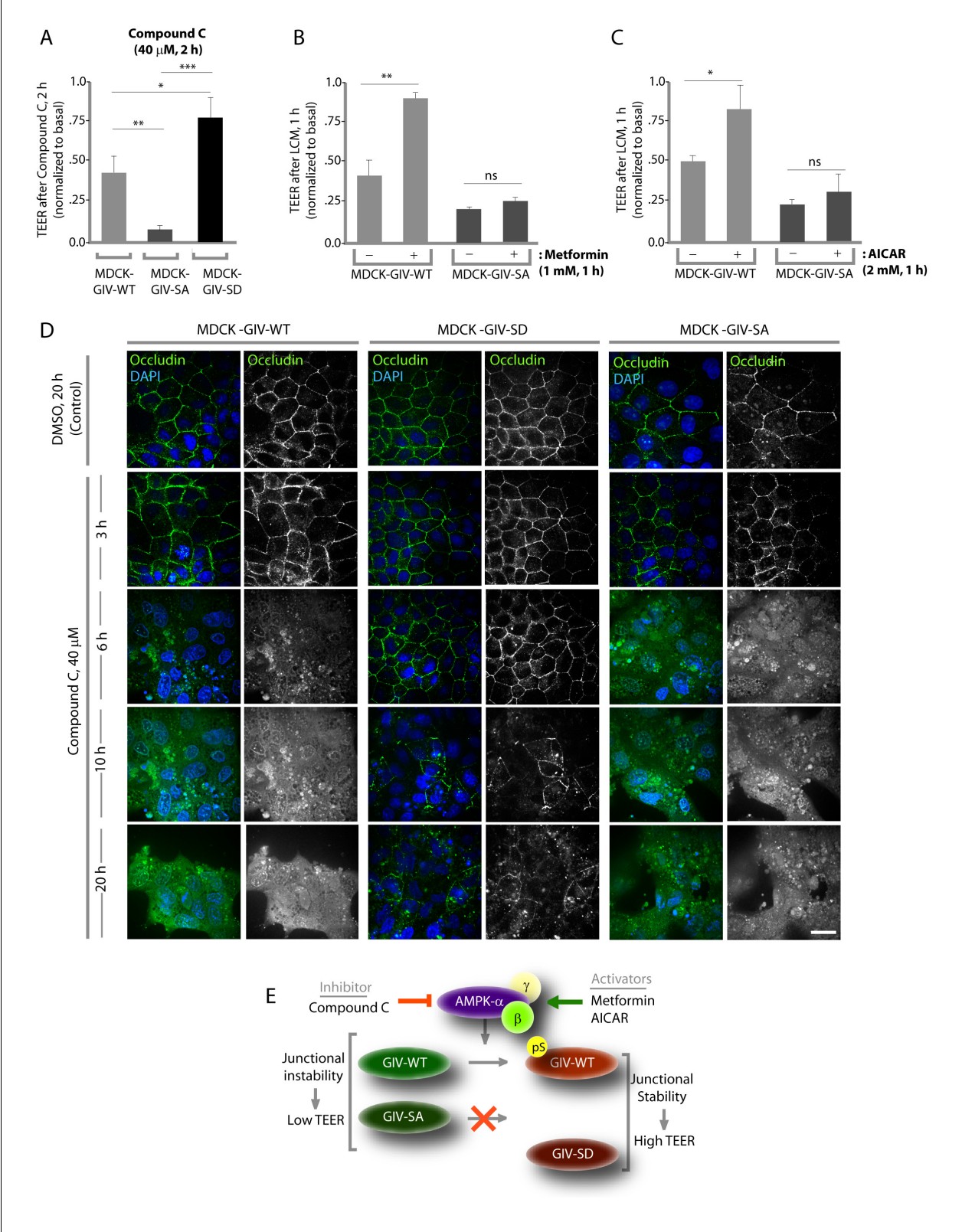

**Figure 5.** Stabilization of tight junctions (TJs) by AMPK requires phosphorylation of GIV at S245. (**A**) MDCK cell lines stably expressing various GIV constructs were grown in Transwell inserts to full confluency into domed monolayers prior to exposing them to Compound C for 1 hr. TEER was measured before and after treatment with Compound C as in 4B-C. Bar graphs display the change in TEER in response to Compound C (compared to basal). MDCK-GIV-SA cells show more drop in TEER and GIV-SD cells show less drop in TEER compared to MDCK-GIV-WT cells. n = 3. Results are

*Figure 5 continued on next page*

*Figure 5 continued*

expressed as ± SEM. *p<0.05; **p<0.01; ***p<0.001. (**B,C**) MDCK-GIV cell lines were grown in Transwell inserts as described above and pre-treated (+) or not (-) for 1 hr with either Metformin (**B**) or AICAR (**C**) prior to exposure to low calcium media (LCM). Bar graphs display the drop in TEER after exposure to LCM. Drop in TEER was significantly less in MDCK-GIV-WT cells when pre-treated with Metformin (**B**) or AICAR (**C**). No significant difference was seen in the degree of drop in TEER in MDCK-GIV-SA cells irrespective of pre-treatment with Metformin (**B**) or AICAR (**C**). n = 3. Results are expressed as ± SEM. ns = not significant; *p<0.05; **p<0.01. (**D**) MDCK cell lines stably expressing various GIV constructs were grown to full confluency into domed monolayers prior to exposing them to Compound C for the indicated periods of time prior to fixation. Fixed cells were stained for the TJ-marker Occludin (green) and DAPI (blue; nuclei) and analyzed by confocal microscopy. Images displayed are representative of 10–11 HPF images captured randomly at 60X mag in each cell line. Loss of TJs, as determined by loss of Occludin staining from cell-cell contact sites, was observed early in MDCK-GIV-SA and WT cells, but preserved for longer in MDCK-GIV-SD cells (compare 6, 10 and 20 hr time points). At each time point, the % area of the monolayer that showed loss of Occludin at cell-cell contact sites was significantly higher in MDCK-GIV-WT (65–90% of the imaged surface area; n = 10 HPFs) and in MDCK-GIV-SA cells (85–100% of the imaged surface area; n = 11 HPFs) when compared to MDCK-GIV-SD cell lines (2–10% of the imaged surface area; n = 10 HPFs; p<0.0001). MDCK-GIV monolayers exposed to Compound C for 20 hr were also monitored by light microscopy, and representative images are shown in *Figure 5—figure supplement 1*. (**E**) Schematic summarizes the effect of the pharmacologic modulators of AMPK used in various panels of this figure and their effect on the AMPK-GIV signaling axis, junctional stability and TEER. Compound C, which inhibits AMPK, inhibits phosphorylation of GIV-WT at S245, destabilizes TJs and reduces TEER. The phosphomimicking GIV-SD mutant which bypasses the need for AMPK, is relatively resistant to the action of Compound C. Metformin and AICAR, two activators of AMPK trigger phosphorylation of GIV-WT at S245, stabilize TJs and preserve TEER. The non-phosphorylatable GIV-SA mutant is relatively insensitive to the actions of Metformin and AICAR.

The following figure supplement is available for figure 5:

**Figure supplement 1.** Phosphorylation of GIV at S245 is sufficient for bypassing the pro-apoptotic actions of the AMPK-inhibitor Coumpound C.

## Phosphorylation at S245 increases GIV's affinity for junction-associated microtubules

Next, we probed the molecular basis for how pS245-GIV stabilizes TJs and resists stress-induced junctional collapse. We hypothesized that phosphorylation at S245 could impact the functions of one of the two N-terminal modules in GIV that flank this residue (*Figure 1A*): (i) the ~200 aa long hook module which binds microtubules (*Simpson et al., 2005*) and (ii) the ~1200 aa long coiled-coil module which mediates homodimerization (*Enomoto et al., 2005*). First, we ruled out a significant impact of phosphorylation at S245 on GIV's ability to homodimerize (*Figure 6—figure supplement 1*). Then, we focused on the impact of such phosphorylation on GIV's ability to associate with microtubules. 3D reconstruction of deconvolved confocal images showed that pS245-GIV colocalized with and followed the bundles of polymerized microtubule tracks at the cell-cell borders (*Figure 6*; *Figure 6—figure supplement 2*). More importantly, pS245-GIV exclusively laced those microtubule tracks that ran along cell-cell junctions (arrowheads; *Figure 6C*) and not those that were found on the free-borders of the cells (arrows; *Figure 6D*). RGB plot analyses revealed two distinct patterns: (1) pS245-GIV overlaying individual microtubule tracks (*Figure 6I*); and (2) pS245-GIV in the middle of two parallel microtubule tracks ('linker' pattern; *Figure 6J*). These findings indicate that pS245-GIV colocalizes with junction-associated microtubules and raise the possibility that the phosphoevent may impact GIV's ability to bind α- and/or β-tubulin heterodimers.

The structure of microtubules shows that the last ~100 amino acids of α- and β-tubulin project from the surface of the MT primarily as two helices (*Nogales et al., 1999*; *Nogales et al., 1998*). We hypothesized that GIV might engage the free C-termini of tubulin monomers, as has been previously demonstrated in the case of the AMPK-substrate CLIP-170 (*Mishima et al., 2007*). We prepared recombinant GST fusion proteins representing the C-terminal ~100 amino acids of α- and β-tubulin (see Materials and methods) and used them in GST pulldown assays. These assays revealed that full length GIV preferentially bound the C-terminus of α-, but not β-Tubulin, and that this interaction is increased during energetic stress (*Figure 7—figure supplement 1*). Using bacterially expressed N-terminal GIV constructs (aa 1–440) that were phosphorylated in vitro with AMPK, we confirmed that the GIV•Tubulin interaction has two key properties: (i) it occurs between GIV's N-terminus and prefers the C-terminus of α-Tubulin (*Figure 7A*); (ii) it requires GIV to be phosphorylated at S245 by AMPK (*Figure 7A–B*). Tubulin co-sedimentation assays further confirmed that energetic stress favors the association of GIV with polymerized, sedimentable microtubules (*Figure 7C–D*), and that phosphorylation at S245 (mimicked by the GIV-SD mutation) is sufficient to enhance such association

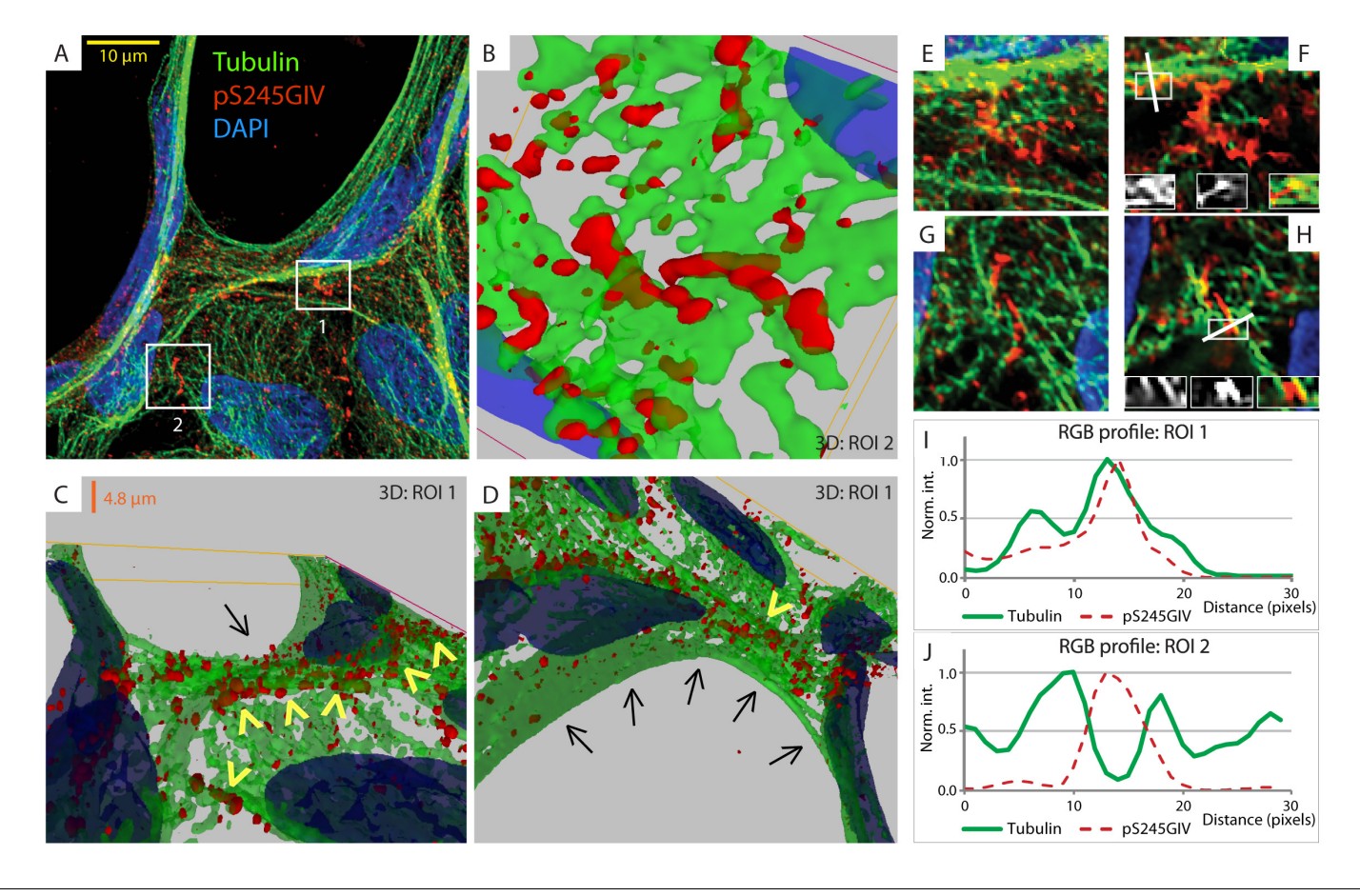

**Figure 6.** GIV phosphorylated at S245 (pS245-GIV) localizes along junction-associated microtubules. (**A**) Subconfluent monolayers of MDCK cells grown on cover slips were fixed and stained for α-Tubulin (green), pS245-GIV (red) and DAPI (blue; nuclei) and analyzed by confocal microscopy. Image **A** shows an xy-plane maximum projection of a deconvolved stack of images, acquired at 20-nm z-intervals using a Olympus cellSens deconvolution (constrained iterative) and restoration system (Olympus FV3000) and a Apochromat 60XOSC2, NA 1.4 objective. The total thickness of the image stack is 4.8 mm. Two regions of interest (ROI), 1 and 2, are indicated with white boxes. (**B–D**) 3D reconstruction of each ROI shows the close proximity of pS245-GIV (red) to connecting microtubules (green) running along the cell-cell junctions (arrowheads). Little or no pS245-GIV staining (red pixels) is seen on microtubules at the 'free' cell border in panel **D** (arrows). (**E–H**) Maximum projected ROI's 1 and 2 are magnified in **E** and **G**, respectively. Individual Z-stacks of each ROI is shown in *Figure 6—figure supplement 2*. Single Z-stacks of ROI's 1 and 2 are displayed in **F** and **H**, respectively. The insets in panels **F** and **H** show red, green and merged channels of respective figures. The white line indicates the pixels used for generating the RGB profile plots shown in **I** and **J**. RGB profiles show that pS245-GIV (red pixels) often co-localizes either completely with microtubule tracks (green pixels; **I**) or lay between two microtubule tracks (**J**) at the cell-cell junctions.

The following figure supplements are available for figure 6:

**Figure supplement 1.** Phosphorylation of GIV at S245 does not affect its ability to homodimerize.

**Figure supplement 2.** GIV phosphorylated at S245 (pS245-GIV) localizes along microtubules at the tight junction (TJ).

(*Figure 7E–F*). These findings demonstrate that phosphorylation at S245 by AMPK impacts GIV's ability to bind α-tubulin and associate with polymerized microtubules. Taken together with our findings by confocal microscopy (*Figure 6*), we conclude that energetic stress triggers pS245-GIV to specifically bind the junction-associated microtubule tracks. Because AMPK regulates post-translational modifications on the C-terminus of α-Tubulin during energetic stress (*Herms et al., 2015*) and because it is capable of stimulating microtubule polymerization specifically at the cell periphery [via phosphorylation of the plus end protein, CLIP-170 (*Nakano et al., 2010*)], it is possible that either or

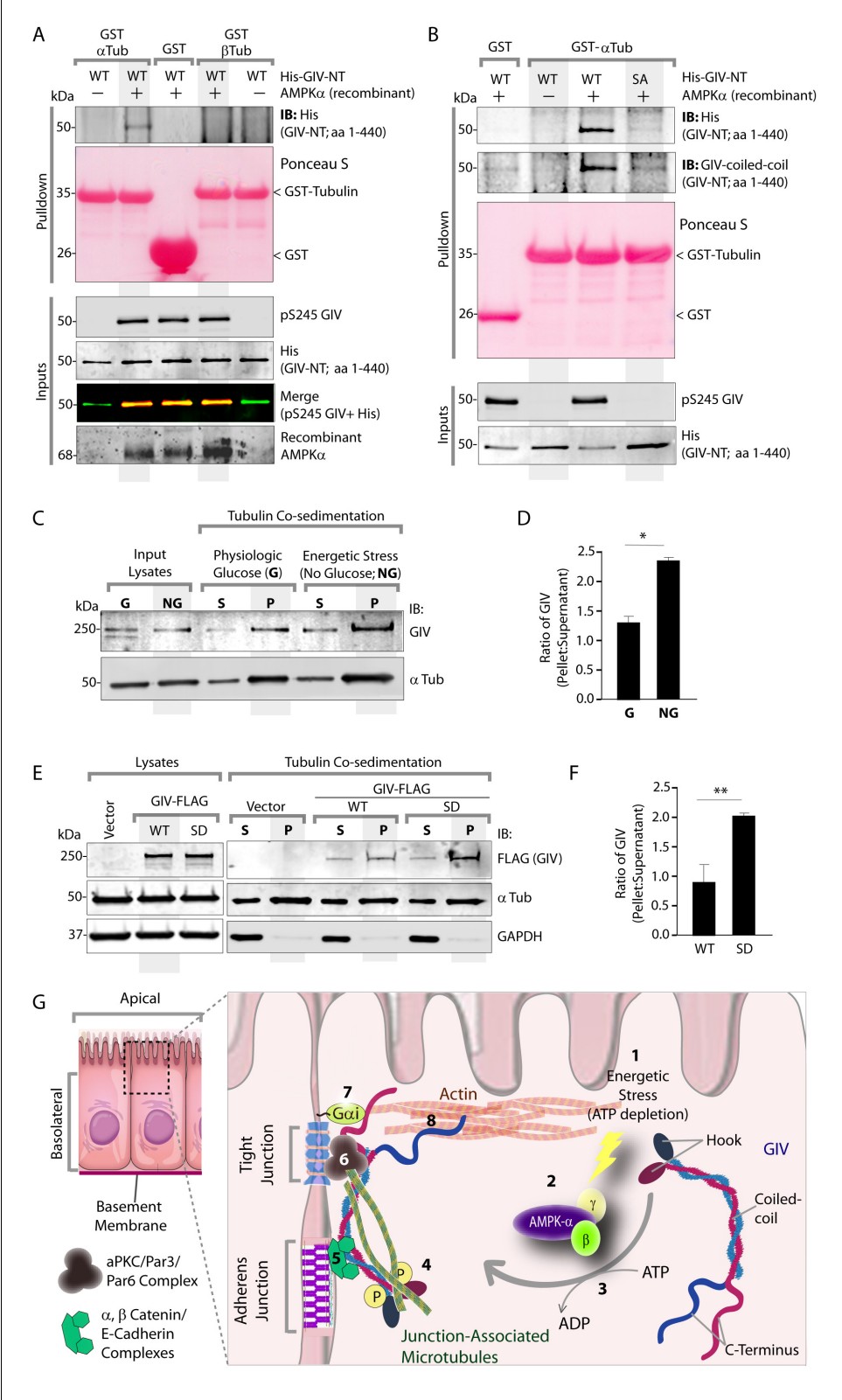

**Figure 7.** Phosphorylation of GIV at S245 increases its ability to bind α-tubulin. (**A**) Bacterially expressed and purified His-GIV-NT WT were phosphorylated (+) or not (-) with recombinant AMPK heterotrimers (α/β/γ) in vitro prior to their use in binding assays with GST-α-Tubulin-CT or GST-β-Tubulin-CT immobilized on glutathione-agarose beads. Bound proteins were analyzed for GIV-NT (anti-His) by immunoblotting (IB). His-GIV-NT(1–440) binds GST-tagged carboxyl terminus of α-Tubulin exclusively after phosphorylation by AMPK. No binding is seen with β-Tubulin regardless of

*Figure 7 continued on next page*

*Figure 7 continued*

phosphorylation. (B) WT and non-phosphorylatable SA mutant His-GIV-NT proteins were phosphorylated (+) or not (-) by recombinant AMPK as in A prior to use in binding assays with GST-tagged carboxyl terminus of α-Tubulin. Bound proteins were analyzed for GIV (His) by immunoblotting (IB). (C–D) Tubulin cosedimentation assays (see Materials and methods) were carried out with equal aliquots of pre-cleared lysates of Cos7 cells that were grown in the presence (G) or absence (NG) of glucose for 18 hr. Samples were centrifuged at high speed to separate the microtubule polymer (P; pellet) from the soluble tubulin (S; supernatant), resolved by SDS-PAGE, and analyzed for the presence of GIV and tubulin by immunoblotting (IB). The ratio of GIV in pellet vs supernatant was quantified by band densitometry and displayed as bar graphs in D. Error bars represent mean ± S.E.M; n = 3; *p<0.05. (E–F) Tubulin cosedimentation assays were carried out with equal aliquots of pre-cleared lysates of Cos7 cells expressing either vector control, or FLAG tagged GIV-WT or SD mutant as in C, and analyzed for the presence of GIV, GAPDH (negative control) and tubulin by immunoblotting (IB). The ratio of GIV in pellet vs supernatant was quantified by band densitometry and displayed as bar graphs in F. Error bars represent mean ± S.E.M; n = 3; **p<0.01. (G) Schematic summarizing the role of GIV in the regulation of cell-cell junction stability during an energetic stress and illustrating how the current findings relate to prior work. Exposure of epithelial cells to conditions that induce energetic stress result in depletion of cellular ATP stores and accumulation of AMP (step 1); the latter activates AMPK kinase (step 2). Once activated, AMPK phosphorylates GIV at S245 (step 3) triggering its localization to the cell-cell junction (TJs) via increased ability to bind TJ-associated microtubules (*Lee et al., 2007*) (step 4). Once localized to the cell-cell junctions, GIV has been shown (*Houssin et al., 2015*) to bind AJ-localized protein complexes, e.g., α- and β-Catenins and E-cadherin and links the catenin-cadherin complexes to the actin cytoskeleton (steps 5 and 8). GIV has also been shown to bind TJ proteins, e.g., aPKC/Par3/Par6 complex (*Ohara et al., 2012*) (step 6), and link these proteins to G proteins and the actin cytoskeleton (*Sasaki et al., 2015*) (steps 7 and 8).

The following figure supplement is available for figure 7:

**Figure supplement 1.** The carboxyl terminus of α-tubulin, but not β-tubulin binds full length GIV from lysates of cells exposed to energetic stress by glucose starvation.

both of those phenomena play a role in restricting pS245-GIV to associate exclusively with the junction-associated microtubule tracks. Although further experiments will be needed to shed light on these mechanisms, the results presented in this work and the published literature to date support the following model (see legend *Figure 7G*): Energetic stress triggers AMPK signaling and phosphorylation of GIV at S245, pS245-GIV localizes to TJs, the N-terminus of GIV binds α-tubulin and junction-associated microtubule tracks. It is here that GIV may subsequently impact cell polarity and junctional integrity by assembling various functional complexes with its C-terminus, i.e., by (i) binding the Par3/Par6/aPKC polarity complex (*Ohara et al., 2012*; *Sasaki et al., 2015*); (ii) binding and modulating the endocytic trafficking of E-cadherin (*Ichimiya et al., 2015*); (iii) linking cadherin-catenin complexes to the actin cytoskeleton (*Houssin et al., 2015*); and finally, (iv) binding and activating G protein, Gαi via its GEF motif and maintaining epithelial polarity through the Par polarity complex (*Sasaki et al., 2015*). Each of these functional associations of GIV have been implicated in the generation of cell polarity.

## The AMPK-GIV axis suppresses anchorage-independent growth; mutations allow some cancer cells to evade such suppression

Next, we asked how the AMPK-GIV stress-polarity pathway impacts cancer cell growth. Most cancers are epithelial in origin, and the loss of cell polarity is a critical step that fuels both malignant transformation and metastatic progression (*Forcet and Billaud, 2007*; *Mirouse and Billaud, 2011*). While activation of AMPK may allow healthy cells to tide over short periods of energetic stress by preserving cell polarity and temporarily restraining energy consumption and growth, tumor cells are known to establish mechanisms to survive prolonged periods of energetic stress by down-regulating AMPK signaling and escaping its restraining influences on growth. To dissect the role of the AMPK-GIV axis, and specifically, pS245-GIV on growth properties of cancer cells, we queried the comprehensive catalogs of genetic mutations identified by genome sequencing, the Cancer Genome Atlas (TCGA) and the Catalogue of Somatic Mutations in Cancer (COSMIC) databases. A search of both databases revealed that the sequence within and flanking the AMPK-consensus site is frequently mutated (*Figure 8—figure supplement 1*). While many of those mutations were predicted by computational modeling to have various degrees of impact on the GIV-AMPKα interaction (see legend, *Figure 8—figure supplement 1*), we chose to characterize one recurrent mutation, L249P (four residues downstream from S245) which was found in two different colorectal tumors. The residue L249 is a key part of the consensus sequence (FxR/KxxS/TxxxL) that is recognized by AMPKα and likely makes a large contribution to the binding energy because it is predicted to be buried in a hydrophobic pocket on

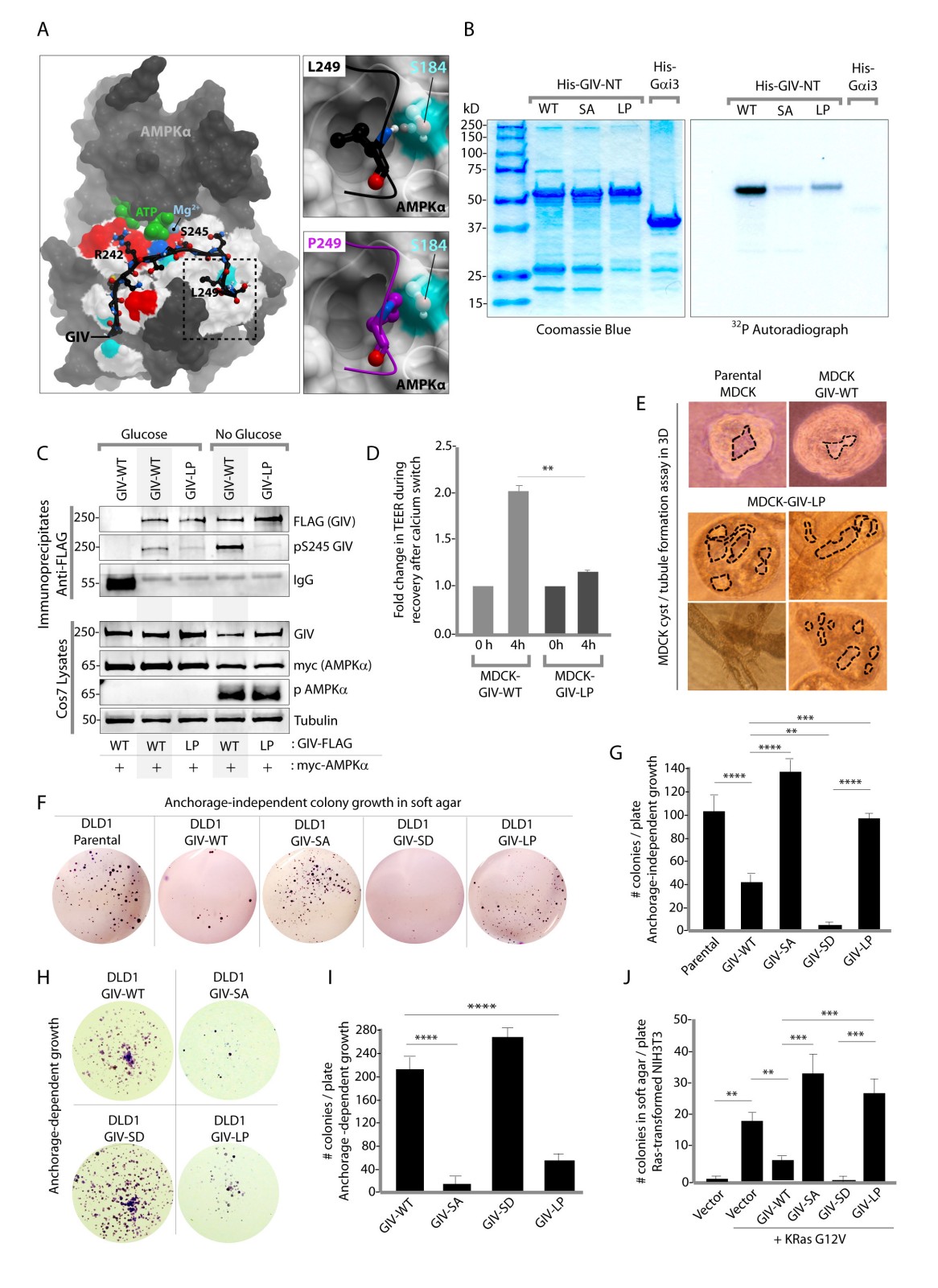

**Figure 8.** An oncogenic non-phosphorylatable GIV mutant reduces junctional stability and favors anchorage-independent growth. (**A**) Homology model of GIV-bound AMPKα is shown on the left. Boxed area in the model is magnified to show the impact of replacing Leu (**L**) at 249 with Pro (**P**). L249 is predicted to be favorably buried in a hydrophobic pocket in AMPKα, contributing to binding energy; P249 is expected to induce a steric clash. See also *Figure 8—figure supplement 1* for a complete list of mutations within and flanking the motif in GIV that is targeted by AMPKα. (**B**) In vitro kinase

*Figure 8 continued on next page*

*Figure 8 continued*

assays were carried out using recombinant AMPK heterotrimers (α/β/γ), bacterially expressed and purified substrates, His-GIV (WT and mutants) and His-Gαi3 (negative control), and γ-32P [ATP]. Phosphoproteins were analyzed by SDS-PAGE followed by autoradiography (right). Equal loading of substrate proteins was confirmed by staining the gel with Coomassie blue (left). AMPK efficiently phosphorylated His-GIV-NT-WT, but not the SA or the LP mutant. (C) In cellulo kinase assays were carried out in Cos7 cells co-expressing GIV-FLAG (WT or LP mutant) and myc-AMPKα constructs, and stimulating AMPK by glucose deprivation for 6 hr prior to lysis. GIV was immunoprecipitated from these lysates using anti-FLAG mAb and analyzed for phosphorylation of GIV at S245 by immunoblotting (IB) with anti-pS245-GIV and FLAG (GIV-Flag). GIV-WT, but not GIV-LP is phosphorylated at S245 in cells responding to energetic stress. (D) Bar graphs display the change in TEER measured across monolayers of MDCK-GIV-WT and LP cells during $Ca^{2+}$ switch. TEER is restored efficiently in MDCK-GIV-WT cells, but not MDCK-GIV-LP cells. (E) MDCK cells grown in collagen containing matrix were analyzed for the formation of cyst and tubular structures by light microscopy as in 5G. Representative cysts and tubular structures are shown. L = lumen. Images were captured at 20X magnification. (F–G) DLD1 colorectal cancer cells stably expressing various GIV constructs were analyzed for their ability to form colonies in soft agar for ~2–3 weeks. Representative fields photographed at 20X magnification are shown in F. Bar graphs in G display the number of colonies (Y axis) seen in each cell line in F, as determined by light microscopy throughout the depth of the matrix in 15 randomly chosen fields. Compared to GIV-WT, GIV-SD inhibits, whereas the non-phosphorylatable GIV-SA and GIV-LP mutants enhance colony formation. (H-I) DLD1 cells in F were analyzed for their ability to form adherent colonies on plastic plates for ~2 weeks prior to fixation and staining with crystal violet. Image of the crystal violet-stained 6-well plate is displayed in H. Bar graphs in I display the number of colonies (Y axis) as determined by ImageJ (colony counter plugin). (J) NIH3T3 cells expressing HA-KRas-G12V alone, or co-expressing HA-KRas-G12V and GIV-WT-FLAG (WT or mutants) were analyzed for their ability to form colonies in soft agar prior to staining with MTT. Bar graphs display the number of colonies formed/plate (Y axis) in each condition. Compared to GIV-WT, GIV-SD inhibits, whereas the non-phosphorylatable GIV-SA and GIV-LP mutants enhance colony formation. n = 3. Results are expressed as ± SEM. ns = not significant; **p<0.01; **p<0.01; ***p<0.001; ****p<0.0001.

The following figure supplement is available for figure 8:

**Figure supplement 1.** The consensus phosphorylation site for AMPK on GIV is frequently mutated in cancers.

AMPKα. Computational modeling analyses predicted that when L249 is substituted with a Proline (P) (*Figure 8A*) a H-bond between L249 and S184 on AMPK will be lost, the Pro will not be able to sit within the hydrophobic pocket where L249 lies, and instead poses a steric clash (see legend; *Figure 8A*). Thus, the L249P mutation (henceforth referred to as LP) was predicted to abolish AMPKα's ability to recognize GIV as a substrate. As predicted, AMPK failed to phosphorylate the GIV-LP mutant as efficiently as GIV-WT either in vitro (*Figure 8B*) or in cells (*Figure 8C*), much like the SA mutant (*Figure 1*). When expressed in MDCK cells, this mutant recapitulated the key phenotypes we observed previously with the non-phosphorylatable GIV-SA mutant, i.e., defective restoration of TJ integrity after $Ca^{2+}$ switch, as determined by measurement of TEER (*Figure 8D*) and defective epithelial morphogenesis, as determined by the formation of abnormal multi-luminal large cysts and abundant tubules in 3D cystogenesis assays (*Figure 8E*). These results demonstrate that the L249P mutant of GIV found in colorectal tumors is not phosphorylatable by AMPK, and its expression in MDCK cells is associated with leakier TJs, disruption of cell polarity and abnormal epithelial morphogenesis.

We hypothesized that GIV-LP may represent a mechanism present in tumor cells that specifically disrupts the AMPK-GIV axis, uncouples energy sensing from cell polarity pathways and thereby, evades the growth restraining effects of AMPK. To test this hypothesis, we generated DLD1 colorectal carcinoma cell lines stably expressing GIV-WT, LP, or the various mutants used in this study and assessed their ability to grow into colonies in a three-dimensional (3D) spatial conformation within soft agar matrix. We chose this cell line because the LP mutant was found in colorectal cancers. The 3D colony-formation assay recreates in culture the typical 3D architecture of the tumor tissue and heterogeneously exposes cancer cells to diffusion-limited glucose, oxygen and nutrients and to other physical and chemical stresses (*Weiswald et al., 2015*) at the tumor core during the early periods of avascular growth. We found that compared to parental DLD1 and DLD1-GIV-WT cells, colony formation was enhanced in those expressing the non-phosphorylatable SA or LP mutants but suppressed in the DLD1-GIV-SD cells (*Figure 8F–G*). These growth patterns were reversed when tumor cells are homogeneously exposed to unlimited oxygen, nutrients, metabolites, and signaling molecules in 2D anchorage-dependent colony growth assays (*Weiswald et al., 2015*) (*Figure 8H–I*); compared to DLD1-GIV-WT cells, DLD1-GIV-SA and LP cells lost the growth advantage previously seen in 3D and showed reduced 2D growth, whereas the DLD-GIV-SD cells overcame the growth suppressive effects previously seen in 3D and showed enhanced 2D growth. These findings indicate that the growth-

enhancing effect of GIV-SA or GIV-LP and the growth-suppressive effect of GIV-SD are restricted to conditions mimicked by anchorage-independent growth in 3D. Those effects were also seen in 3D growth of Ras-transformed NIH3T3 fibroblasts (*Figure 8J*), indicating that the AMPK-GIV axis may in part be responsible for mediating the previously observed growth suppressive action of AMPK in this model system (*Ng et al., 2012*; *Phoenix et al., 2012*). These results indicate that disruption of the AMPK-GIV signaling axis (mimicked by GIV-SA or LP) favors anchorage-independent 3D growth, whereas constitutive activation of the pathway (mimicked by GIV-SD) prevents such growth. We conclude that AMPK-dependent phosphorylation of GIV at S245 restricts epithelial cell growth to anchorage-dependent conditions via its ability to protect the epithelium from stress-induced junctional collapse, but fails to grow without anchorage, presumably because of death (anoikis) when anchorage is lost. Tumor cells expressing GIV mutants that cannot be phosphorylated by AMPK evade this protective mechanism.

## AMPK agonists exert their tumor suppressive effects via the AMPK-GIV axis

AMPK agonists such as Metformin and AICAR inhibit anchorage-independent 3D growth of a variety of cancer cells (*Vincent et al., 2015*), e.g., lung, pancreatic, colon, ovarian, breast, prostate, renal cancer cells, melanoma, and even acute lymphoblastic leukemia cells [reviewed in (*Shackelford and Shaw, 2009*)]. We asked if the growth suppressive action of AMPK agonists requires an intact AMPK-GIV signaling axis. Anchorage-independent growth assays were carried out using various DLD1-GIV cell lines in the presence of increasing doses of AICAR or Metformin. As expected, both AMPK agonists effectively suppressed the growth of parental DLD1 and DLD1-GIV-WT cells in a dose-dependent manner (*Figure 9A–B*), whereas the growth of DLD1-GIV-SA and LP cells was only mildly suppressed and only at higher doses. These findings indicate that phosphorylation of GIV at S245 by AMPK is essential for the growth suppressive actions of AMPK agonists. We conclude that anti-proliferative actions of AMPK agonists like Metformin is mediated at least in part via the AMPK-GIV signaling axis. These findings, along with the findings in *Figures 4–8* support the following model: In normal epithelial cells in which the AMPK-GIV axis is intact (*Figure 9C*) pS245-GIV serves as a major effector of AMPK at the TJs, inhibits stress-induced junctional collapse, preserves cell polarity, and the epithelial barrier functions to tide over stress events. Under these conditions, AMPK agonists restrict epithelial cell growth to anchorage-dependent conditions. In tumor cells in which the AMPK-GIV axis is disrupted (*Figure 9D*) TJs are vulnerable to stress-induced collapse, cells lose polarity and overcome the need for anchorage for growth. Under these conditions, anchorage-independent growth is favored, which initiates and sustains cancer growth, and AMPK agonists are rendered ineffective.

## Conclusions

Findings showcased in this work help draw three major conclusions. The first conclusion is that the polarity scaffold protein GIV is a direct target and an effector of the energy-sensing kinase AMPK; phosphorylation of GIV at a single site is necessary and sufficient to mediate the junction-stabilizing functions of AMPK in the normal epithelium. In doing so, this work has revealed an elusive link between the stress-sensing components and the cell polarity pathways (see legend, *Figure 10*), and improved our understanding of molecular mechanisms of how epithelial monolayers are protected despite being constantly bombarded by energetic stressors by fortifying cell-cell junctions against stress-induced collapse. Because GIV serves as a junctional scaffold in both epithelial (*Sasaki et al., 2015*) and endothelial cells (*Ichimiya et al., 2015*), it is possible that the stress-triggered mechanisms outlined here also account for the protective role of AMPK on cell junctions observed consistently in a variety of tissue and cell types, in both epithelial (*Garnett et al., 2013*; *Patkee et al., 2016*; *Seo-Mayer et al., 2011*; *Spruss et al., 2012*) and endothelial (*Castanares-Zapatero et al., 2013*; *Liu et al., 2014*; *Takata et al., 2013*) cells, in the face of diverse chemical, bacterial and metabolic stressors. Given the overlapping substrate specificity of AMPK and its related kinases [reviewed in (*Shackelford and Shaw, 2009*)], it seems likely that AMPK-related family members such as the MARKs, may phosphorylate S245 on GIV under other conditions or in specific tissues. Because GIV is a GEF for trimeric G proteins (*Garcia-Marcos et al., 2009*), and because GIV's ability to trigger Gαi signaling is essential for cell polarity (*Sasaki et al., 2015*), it is possible that localized GIV-dependent

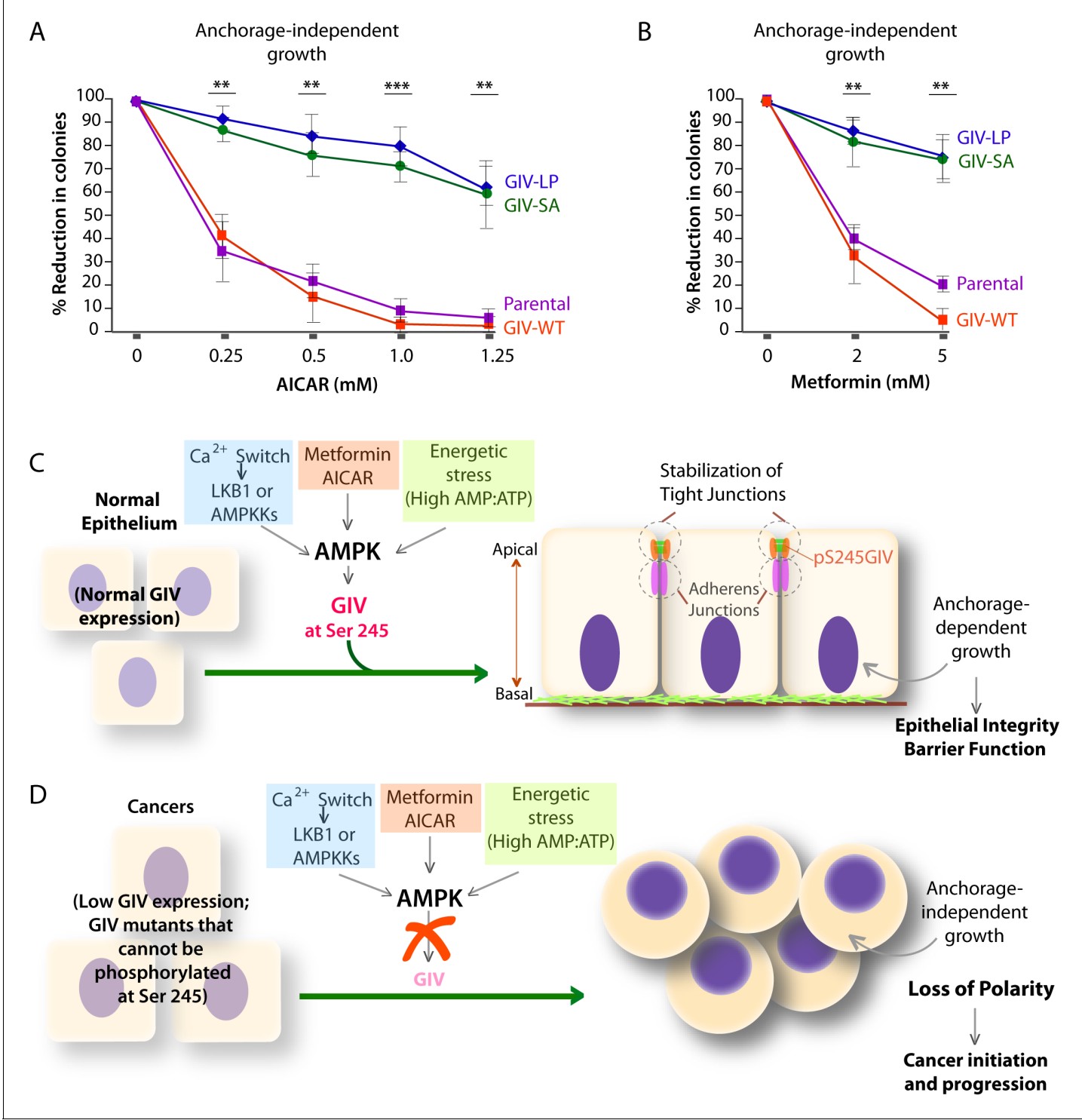

**Figure 9.** Phosphorylation of GIV at S245 is essential for the tumor-suppressive actions of AICAR and Metformin. (**A**, **B**) Parental DLD1 colorectal cancer cells and those stably expressing GIV-WT and the non-phosphorylatable SA and LP mutant GIV constructs were analyzed for their ability to form colonies in soft agar for 2–3 weeks in the presence of increasing concentrations of AICAR (**A**) or Metformin (**B**). The number of colonies was counted by light microscopy throughout the depth of the matrix in 15 randomly chosen fields. Graphs display the % reduction in colonies (Y axis) in each cell line, normalized to their respective controls (no treatment). Error bars represent mean ± S.E.M; n = 3; **p<0.01; ***p<0.001; ****p<0.0001. The growth suppressive effects of AICAR (**A**) and Metformin (**B**) observed in DLD1 cells expressing GIV-WT were significantly reduced in cells expressing GIV-SA or GIV-LP. (**C**,**D**) Working models for how the AMPK-GIV axis impacts cell polarity in normal epithelial cells and in cancers. In the presence of a functional AMPK-GIV axis (**C**) in which GIV can be phosphorylated by AMPK at S245, cell polarity and the integrity of TJs are preserved in the face of energetic

*Figure 9 continued on next page*

*Figure 9 continued*

stress, thereby maintaining epithelial barrier function and homeostasis and anchorage-dependency for growth. In the absence of a functional AMPK-GIV axis (D), either because AMPK cannot be activated or GIV cannot be phosphorylated by AMPK at S245, TJs succumb to energetic stress, cell polarity is lost, and cells can undergo anchorage-independent sustained growth.

G-protein signaling at the cell-cell junctions triggered by energetic stress may modulate cellular stress response via G-protein intermediates.

The second conclusion is that the AMPK-GIV stress-polarity pathway inhibits oncogenic transformation and growth and that disruption of this pathway (accomplished via mutations identified during genomic sequencing of colorectal cancers) helps tumor cells escape such inhibition and gain proliferative advantage during 3D growth. These conclusions are in keeping with prior observations that polarity defects precede the onset of tumorigenesis when the LKB1-AMPK pathway is inhibited (*Hezel et al., 2008*), fueling the speculation that polarity defects may be one of the major mechanisms for tumor initiation when the energy sensing pathway is dysregulated (*Hezel and Bardeesy, 2008*). Multiple studies on GIV have demonstrated its role as a bona fide metastasis-related protein across a variety of cancers, primarily via its ability to initiate and amplify tyrosine-based G protein signals (*Aznar et al., 2016*; *Ghosh, 2016*) and couple it to actin cytoskeletal remodeling at the leading edge of migrating cells predominantly via interactions mediated by its C terminus [reviewed in (*Ghosh, 2015*)]. This study is the first of its kind that reveals the protective role of GIV's N terminus. Because loss of cell polarity is known to impact both early and later steps of oncogenic progression (*Wodarz and Nathke, 2007*), it is possible that loss of cell polarity due to the disruption of the AMPK-GIV axis impacts not just the early phases of transformation and cancer growth, but also fuels cancer invasion into adjacent tissues and the formation of metastases.

The third conclusion is that the AMPK-GIV axis appears to be necessary for at least some of the protective actions of the AMPK agonist Metformin on the epithelium. Metformin (Glucophage), the most widely used type II diabetes drug in the world, reduces blood glucose by activating the LKB1-AMPK pathway (*Shaw et al., 2005*) and inhibiting hepatic gluconeogenesis [reviewed in (*Shackelford and Shaw, 2009*)]. Besides its ability to lower blood glucose, Metformin also exerts two other effects in an AMPK-dependent manner: (i) it stabilizes cell-cell junctions and protects barrier functions of both epithelial and endothelial monolayers in the setting of a variety of pathologic stressors; and (ii) it suppresses the growth of a variety of tumor cells and embryonic stem cells in culture and tumor xenografts in mice [reviewed in (*Shackelford and Shaw, 2009*)]. By demonstrating that phosphorylation of GIV by AMPK is required for Metformin to exert both these effects efficiently, this work unravels that activation of AMPK, and more specifically, the AMPK-GIV signaling axis as an important mechanism of action of Metformin. It is noteworthy that although multiple retrospective clinical trials have generally concluded that prolonged use of Metformin reduces the incidence of cancer, others have reported conflicting results, and several prospective clinical trials are underway to identify which target populations may specifically benefit from this drug [reviewed in (*Kourelis and Siegel, 2012*; *Pryor and Cabreiro, 2015*)]. Given the widespread long-term use of metformin as a prescription drug and its potential utility both in chemoprevention as well as chemotherapy, further studies are warranted to investigate if the GIV-expression status in tumors [e.g., its expression as a spliced isoform lacking the C terminus (*Ghosh et al., 2010*) or mutants that prevent phosphorylation by AMPK (current work), or its overexpression as full length (*Ghosh et al., 2016*)] may help identify which patients may benefit from the tumor suppressive actions of the Metformin.

In summary, this study not only populates the stress-polarity pathway with a new player, i.e., GIV, and reveals how this player and the pathway may be altered in cancers, but also it provides mechanistic insights into the epithelium-protecting and tumor-suppressive actions of one of the most widely prescribed drugs, the AMPK-agonist Metformin.

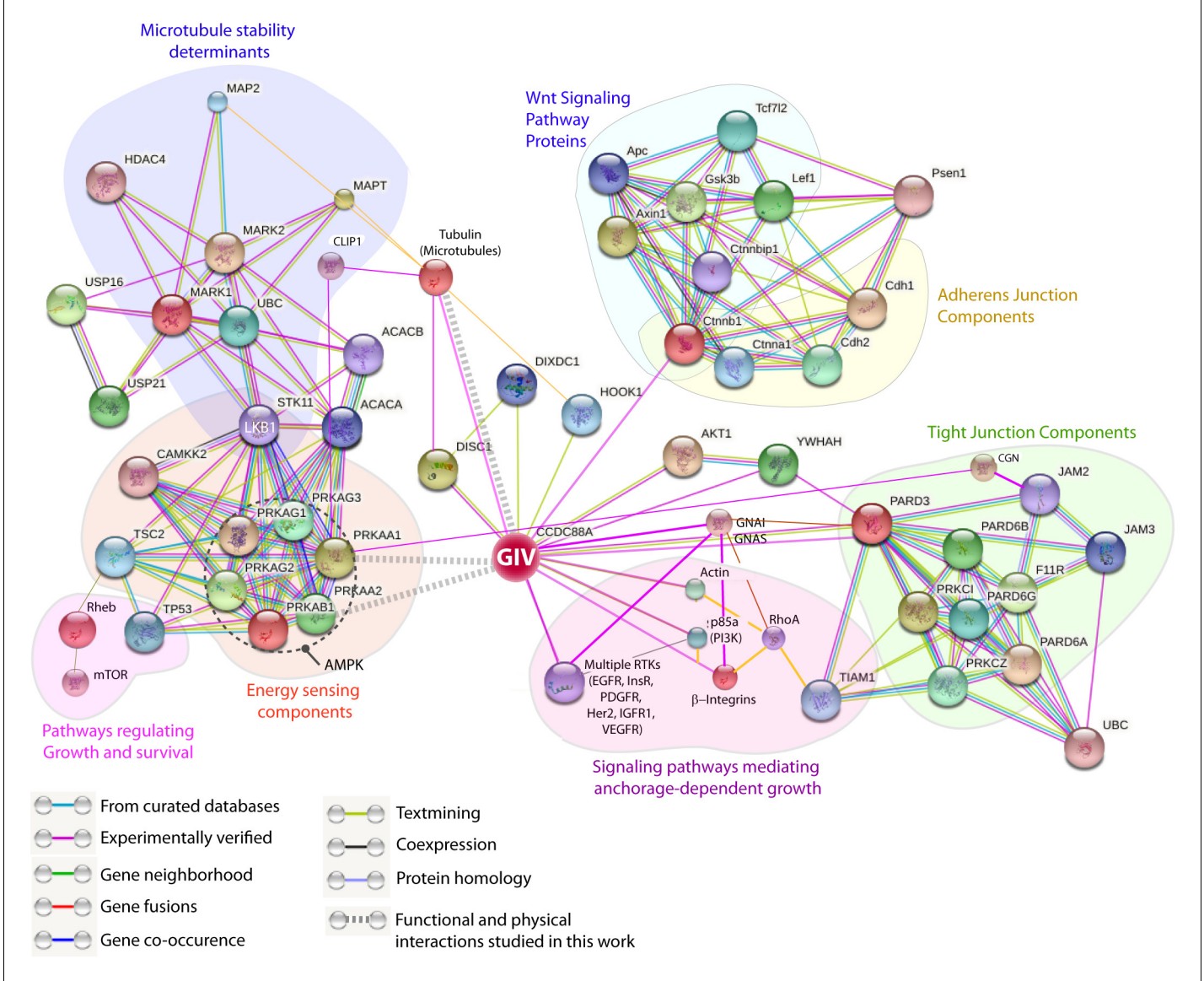

**Figure 10.** GIV (CCDC88A) functions at the nexus of a variety of cellular pathways that regulate and respond to changes in cell-cell junctions. A protein-protein interaction network (see Materials and methods) shows all known and predicted protein-protein interactions (relevant to the current work), both functional and directly physical in nature, that stem from computational prediction, from knowledge transfer between organisms, and from interactions reported in other (primary) databases. The clusters of functional associations shown here are meant to be specific and meaningful, i.e. proteins jointly contribute to a shared function; this does not necessarily mean they are physically binding each other. The interaction network shows that GIV (CCDC88A) functions as a key polarity-determinant scaffold which links major components that sense cellular energy (STK11, LKB1; PRKA, AMPK) and determine cellular growth and survival responses (on the left) and signaling pathways, e.g., Wnt (top; light blue) and growth factor and integrin pathways (purple; bottom) that dictate anchorage-dependent vs independent growth in normal and transformed cells, respectively, to cell-cell junctions (both tight and adherens; on the right). The microtubule stability determinants (upper left; dark blue) are likely to synergize with the AMPK-GIV axis for protecting junctions against stress-induced collapse. Two previously defined functional interactions of PRKA (AMPK) studied in different contexts stand out as key links: (1) Phosphorylation of the microtubule plus end protein CLIP-170 (CLIP1; deep blue) by AMPK enhances the speed of microtubule polymerization (*Nakano et al., 2010*); and (2) Phosphorylation of the TJ- protein cingulin (CGN; green) by *AMPK* increases the association of plus ends of microtubules with TJs (*Yano et al., 2013*). Key at the lower left corner indicates how each functional interaction within this network was color-coded.

## Materials and methods

### Reagents and antibodies

Unless otherwise indicated, all reagents were of analytical grade and obtained from Sigma-Aldrich (St. Louis, MO). Cell culture media were purchased from Invitrogen (Carlsbad, CA). All restriction endonucleases and Escherichia coli strain DH5α were purchased from New England Biolabs (Ipswich, MA). *E. coli* strain BL21 (DE3), phalloidin-Texas Red were purchased from Invitrogen. Genejuice transfection reagent was from Novagen (Madison, WI). PfuUltra DNA polymerase was purchased from Stratagene (La Jolla, CA). Goat anti-rabbit and goat anti-mouse Alexa Fluor 680 or IRDye 800 F (ab')$_2$ used for immunoblotting were from Li-Cor Biosciences (Lincoln, NE). Mouse anti-His, anti-FLAG (M2), anti-α tubulin and anti-actin were obtained from Sigma; anti-Myc was obtained from Cell Signaling (Danvers, MA); anti-Occludin and anti-ZO1 were obtained from Zymed (San Francisco, CA); anti-β-catenin and E-cadherin were obtained from Santa Cruz Biotechnology (Dallas, TX). Rabbit anti-Gαi3 and anti-GIV CT were obtained from Santa Cruz Biotechnology; anti-GIV-coiled-coil (CC) was obtained from EMD Millipore (Carlsbad, CA), anti-AMPK and anti-pAMPK were obtained from Cell Signaling. The affinity purified anti-pS245-GIV rabbit polyclonal antibody was generated using a phosphopeptide corresponding to the sequence flanking S245 in human GIV as immunogen by 21st Century Biochemicals (Marlboro, MA).

### Plasmid constructs and mutagenesis

For mammalian expression, a well-characterized and extensively validated C-terminal FLAG-tagged construct (*Ghosh et al., 2010*) was used. It was originally generated by cloning human GIV (NCBI RefSeq Accession: Q3V6T2) into p3XFLAG-CMV-14 between *Not*I and *Bam*HI. All subsequent site-directed mutagenesis (GIV-Flag full length S245A (SA), S245D (SD) and L249P (LP) were carried out on this template using Quick Change as per the manufacturer's protocol. The GST-GIV-NT WT, SA and LP (1–440 aa), GST-GIV-NT WT (440-747aa), His-GIV WT, SA, SD and LP constructs (1–440 aa) used for in vitro protein-protein interaction and kinase assays were cloned from GIV-FLAG p3XFLAG-CMV-14 and inserted within the pGEX-4T or pET28b vectors, respectively, between *Nde*I/*EcoR*I restriction sites. Previously validated HA-Ras G12V (*Aznar et al., 2015*) and pcDNA3-Myc-tagged wild-type mouse AMPKα$_2$ (*Mu et al., 2001*) constructs were generous gifts from Robert Hayward (King's College London, London, UK) and Morris Birnbaum (University of Pennsylvania, Philadelphia, USA). Previously validated GST tagged carboxyl termini of human α1A- (aa 399–451) and β2B- (aa 390–445) tubulins in the pET49b vector backbone were generous gifts from Toshio Hakoshima (Nara Institute of Science Technology, Nara, Japan) (*Hirano et al., 2011*).

### Construction of a homology model of GIV bound to the catalytic α-subunit of AMPK

The homology model of the AMPKα•GIV complex was constructed using the ICM comparative (homology) modeling procedure (*Cardozo et al., 1995*) using the structure of constitutively active Par1-MARK2 (a member of the AMPK family of kinases) in complex with the CagA protein encoded by pathologic strains of *Helicobacter pylori* [PDB: 3IEC (*Nesic et al., 2010*)] as a template and guided by the GIV/AMPKα sequence alignment in *Figure 1A*. The initial model was by assigning the backbone coordinates of both target molecules (AMPKα and GIV) to their counterparts in the template; this model was further refined using extensive sampling of residue side chains in internal coordinates and then additionally relaxed by full-atom local minimization in the presence of distance restraints maintaining the conserved hydrogen bonds and thus protein secondary structure and topology.

### Cell culture and the rationale for choice of cells in various assays

Tissue culture was carried out essentially as described before (*Garcia-Marcos et al., 2011a*; *Ghosh et al., 2008*, *2010*). We used a total of five different cell lines in this work, each chosen carefully based on its level of endogenous GIV expression and the type of assay. AMPK+/+ and AMPK-/- SV40-immortalized MEFs that have been extensively validated in prior publications (*Lamia et al., 2009*) were a generous gift from Katja Lamia (SALK Institute, La Jolla) with permission from Benoit Viollet (Institut Cochin INSERM, France). Parental Cos7, MDCK, NIH3T3 and DLD1 cell lines were

obtained from ATCC and cultured at 37°C in a 5% CO2 humidified incubator according to the ATCC guidelines. Briefly, Cos7, NIH3T3 and MDCK cells were grown in Dulbecco's modified Eagle's medium (DMEM) supplemented with 10% fetal bovine serum (FBS) and 1% penicillin-streptomycin-glutamine. DLD1 cells were grown in Roswell Park Memorial Institute medium (RPMI) supplemented with 10% fetal bovine serum (FBS) and 1% penicillin-streptomycin-glutamine. Cells were passaged at 90–100% confluency using ATV. All these cell lines were cultured according to the ATCC guidelines, screened for drifting by short tandem repeat (STR) profiling services offered by ATCC for all lines >10 passages, and periodically monitored for mycoplasma infection (once a year).

Cos7 cells were primarily used to transiently overexpress tagged proteins for use in biochemical studies; lysates of these cells were used as source of proteins in various protein-protein interaction (immunoprecipitation and pulldown) and in cellulo kinase assays. We use Cos7 cells because they are easy to transfect (> 90% efficiency) with most constructs.

Type II MDCK cells were primarily used to study the effect of GIV on cell polarity. These cells have been extensively characterized in 2D and 3D cultures to study cell-cell junctions and cell polarity integrity. We determined that levels of GIV are significantly lower (~10 fold) in these cells compared to MDCK type I cells, thereby allowing us to reconstitute GIV expression exogenously and analyze the effect of various mutant GIV constructs without significant interference due to the endogenous protein.

DLD1 cells were primarily used to investigate the role of the AMPK-GIV axis on growth properties (anchorage-dependent and independent) of colorectal cancer cells. There are several reasons why this colorectal cancer cell line was chosen: (1) We focused on colorectal cancer in this study because the oncogenic mutant we characterized was found in two different colorectal tumors. Thus, DLD1 cells were deemed appropriate for us to translate our findings. (2) These cells have been extensively characterized with respect to most oncogenes (ATCC database), and are highly tumorigenic in 2D and 3D cultures due to a mutation in KRAS (G13D) (*Ahmed et al., 2013*; *Shirasawa et al., 1993*).

Low-passage NIH3T3 fibroblasts were used exclusively in neoplastic transformation assays to study the role of the AMPK-GIV axis during neoplastic transformation. Ras-transformed NIH3T3 cells were used because this is the gold standard assay used to study the role of a gene/protein in tumor transformation (*Egan et al., 1987*). Like Cos7 cells, these cells are also highly transfectable (~80% transfection efficiency with GIV-FLAG). Such ease of co-expression, consistently at high levels allows us to study the effect of various mutant GIV constructs on Ras-transformed growth.

## Generating bioenergetic stress

To activate AMPK in various assays, we subjected cells to energetic stress (mimic ATP-depleted state with high AMP/ADP to ATP ratios) by exposing them to glucose-free media for indicated durations as done previously by others [reviewed in (*Hardie, 2011*)]. Whenever such energetic stress was used in various assays, activation of AMPK was confirmed by immunoblotting in each instance. Besides glucose deprivation, we also eliminated growth factors/serum in these assays. This is because GIV has been previously implicated as an upstream modulator of growth factor triggered metabolic response, e.g., its ability to modulate mTOR activation (*Garcia-Marcos et al., 2011a*; *Wang et al., 2015*), regulate the exocytosis of glucose transporters and uptake of glucose into cells responding to insulin (*Lopez-Sanchez et al., 2015*; *Ma et al., 2015*), as well as its ability to trigger reversal of autophagy in response to growth factors (*Garcia-Marcos et al., 2011a*). We carried out all energetic stress assays in the absence of serum (which is a source of multiple growth factors) so as to avoid unanticipated and confounding feedback loops (from growth factor receptors to AMPK via GIV) from making interpretation difficult in assays designed to selectively dissect the role of signaling from AMPK to GIV.

## Transfection, generation of stable cell lines and cell lysis

Transfection was carried out using Genejuice (Novagen) for DNA plasmids following the manufacturers' protocols. MDCK and DLD1 cell lines stably expressing GIV constructs were selected after transfection in the presence of 800 μg/ml G418 for 6 weeks. The resultant multiclonal pool was subsequently maintained in the presence of 500 μg/ml G418. GIV expression was verified independently by immunoblotting using anti-Flag and anti-GIV antibodies, and estimated to be ~3x the endogenous level. Unless otherwise indicated, for assays involving glucose starvation, cells were incubated

with DMEM without serum and glucose. Cells were treated with 4 mM EGTA (Fisher Scientific; Waltham, MA) in the appropriate experiments.

Whole-cell lysates were prepared after washing cells with cold PBS prior to resuspending and boiling them in sample buffer. Lysates used as a source of proteins in immunoprecipitation or pull-down assays were prepared by resuspending cells in Tx-100 lysis buffer [20 mM HEPES, pH 7.2, 5 mM Mg-acetate, 125 mM K-acetate, 0.4% Triton X-100, 1 mM DTT, supplemented with sodium orthovanadate (500 µM), phosphatase (Sigma) and protease (Roche) inhibitor cocktails], after which they were passed through a 28G needle at 4°C, and cleared (10,000 x g for 10 min) before use in subsequent experiments.

## Protein expression and purification

GST and His-tagged recombinant proteins were expressed in *E. coli* strain BL21 (DE3) (Invitrogen) and purified as described previously (*Garcia-Marcos et al., 2011a*; *Ghosh et al., 2010*; *Ghosh et al., 2008*). Briefly, bacterial cultures were induced overnight at 25°C with 1 mM iso-propylβ-D-1-thio-galactopyranoside (IPTG). Pelleted bacteria from 1 L of culture were resuspended in 20 mL GST-lysis buffer [25 mM Tris·HCl, pH 7.5, 20 mM NaCl, 1 mM EDTA, 20% (vol/vol) glycerol, 1% (vol/vol) Triton X-100, 2X protease inhibitor mixture (Complete EDTA-free; Roche Diagnostics)] or in 20 ml His-lysis buffer [50 mM NaH2PO4 (pH 7.4), 300 mM NaCl, 10 mM imidazole, 1% (vol/vol) Triton X-100, 2X protease inhibitor mixture (Complete EDTA-free; Roche Diagnostics)] for GST or His-fused proteins, respectively. After sonication (three cycles, with pulses lasting 30 s/cycle, and with 2 min intervals between cycles to prevent heating), lysates were centrifuged at 12,000X g at 4°C for 20 min. Except for GST-α/β Tubulin-CT constructs (see in vitro GST pulldown assay section), solubilized proteins were affinity purified on glutathione-Sepharose 4B beads (GE Healthcare) or His-Pur Cobalt Resin (Pierce), dialyzed overnight against PBS, and stored at −80°C.

## In vitro GST pulldown and immunoprecipitation assays

Purified GST-GIV-NT or GST alone (5 µg) were immobilized on glutathione-Sepharose beads and incubated with binding buffer [50 mM Tris-HCl (pH 7.4), 100 mM NaCl, 0.4% (v:v) Nonidet P-40, 10 mM MgCl$_2$, 5 mM EDTA, 30 µM GDP, 2 mM DTT, protease inhibitor mixture] for 90 min at room temperature as described before (*Garcia-Marcos et al., 2011a*; *Ghosh et al., 2010*; *Ghosh et al., 2008*; *Lin et al., 2011*). Lysates (~250 µg) of Cos7 cells expressing AMPK construct were added to each tube, and binding reactions were carried out for 4 hr at 4°C with constant tumbling in binding buffer [50 mM Tris-HCl (pH 7.4), 100 mM NaCl, 0.4% (v:v) Nonidet P-40, 10 mM MgCl$_2$, 5 mM EDTA, 30 µM GDP, 2 mM DTT]. Beads were washed (4X) with 1 mL of wash buffer [4.3 mM Na$_2$HPO$_4$, 1.4 mM KH$_2$PO$_4$ (pH 7.4), 137 mM NaCl, 2.7 mM KCl, 0.1% (v:v) Tween 20, 10 mM MgCl$_2$, 5 mM EDTA, 30 µM GDP, 2 mM DTT] and boiled in Laemmli's sample buffer. Immunoblot quantification was performed by infrared imaging following the manufacturer's protocols using an Odyssey imaging system (Li-Cor Biosciences).

GST- α/β Tubulin-CT constructs were immobilized on glutathione-Sepharose beads directly from bacterial lysates by overnight incubation at 4°C with constant tumbling. The next morning, GST-α or β Tubulin-CT immobilized on glutathione beads were washed and subsequently incubated with appropriate His-tagged GIV-NT constructs at 4°C with constant tumbling. Washes and immunoblotting were performed as previously.

For immunoprecipitation studies, cell lysates (~1–2 mg of protein) were incubated for 4 hr at 4°C with 2 µg of appropriate antibody, anti-Flag mAb (Sigma Aldrich) for GIV-FLAG, anti-Myc mAb (Cell Signaling) for Myc-AMPK, anti-GIV-CT Ab (SCBT) for endogenous GIV or their respective pre-immune control IgGs. Protein G (for all mAbs) or protein A (for polyclonal anti-GIV Ab) Sepharose beads (GE Healthcare) were added and incubated at 4°C for an additional 60 min. Beads were washed in PBS-T buffer [4.3 mM Na$_2$HPO$_4$, 1.4 mM KH$_2$PO$_4$, pH 7.4, 137 mM NaCl, 2.7 mM KCl, 0.1% (v:v) Tween 20, 10 mM MgCl$_2$, 5 mM EDTA, 2 mM DTT, 0.5 mM sodium orthovanadate], and bound proteins were eluted by boiling in Laemmli's sample buffer.

## In vitro and in cellulo kinase assays

In vitro radioactive kinase assays were carried out as described previously (*Bhandari et al., 2015*) using purified His- or GST- tagged GIV-NT proteins (WT, L249P and S245A mutant; 3–5 µg/reaction)

and recombinant, purified AMPKα2 heterotrimers (Signal Chem; 100 ng/reaction). The reactions were started by addition of ATP (5 µCi/reaction γP32-ATP and 6 µM cold ATP) and carried out at 30°C in 30 µl of kinase buffer (20 mM Tris·HCl, pH 7.5, 2 mM EDTA, 10 mM MgCl2, 0.1 mM AMP and 1 mM DTT). Phosphorylated proteins were separated on 10% SDS-PAGE and detected by auto-radiography. Non-radioactive in vitro kinase assays were carried out as described above, except the radioactive ATP mix was replaced with cold ATP (25 µM) in 30 µl of kinase buffer. Phosphorylated proteins were separated on 10% SDS-PAGE and detected by immunoblotting with anti-pS245-GIV antibody.

In cellulo kinase assays were carried out in Cos7 cells co-expressing both substrate and kinase proteins as done previously (*Lin et al., 2011*). Briefly, myc-tagged AMPKα2 and GIV-FLAG (WT and mutant) constructs were co-expressed in Cos7 cells, and the kinase was activated by subjecting the cells to energetic stress by glucose-starvation overnight (16–18 hr) prior to lysis. GIV was immuno-precipitated from these lysates and analyzed for phosphorylation at S245 by immunoblotting with anti-pS245-GIV antibody.

## Quantitative immunoblotting

For immunoblotting, protein samples were separated by SDS-PAGE and transferred to PVDF membranes (Millipore). Membranes were blocked with PBS supplemented with 5% nonfat milk (or with 5% BSA when probing for phosphorylated proteins) before incubation with primary antibodies. Infrared imaging with two-color detection and band densitometry quantifications were performed usinga Li-Cor Odyssey imaging system exactly as done previously (*Garcia-Marcos et al., 2010* ;*2011a*; *2011b*; *2012*;*Ghosh et al., 2010*) All Odyssey images were processed using ImageJ software (NIH) and assembled into figure panels using Photoshop and Illustrator software (Adobe).

## Immunofluorescence

MDCK cell lines were fixed at room temperature with 3% paraformaldehyde for 20–25 min, permea-bilized (0.2% Triton X-100) for 45 min and incubated for 1 hr each with primary and then secondary antibodies as described previously (*Ghosh et al., 2008*). Dilutions of antibodies and reagents were as follows: anti-GIV (1:300); anti-phospho-Ser245-GIV (pS245-GIV; 1/250); anti-Occludin (1/250); anti-ZO-1 (1/250); anti-β-catenin (1/250); anti-E-cadherin (1/250); Phalloidin (1:1000); DAPI (1:2000); goat anti-mouse (488 and 594) Alexa-conjugated antibodies (1:500). Images were acquired using a Leica CTR4000 Confocal Microscope with a 63X objective. Z-stack images were obtained by imaging approximately 4-µm thick sections of cells in all channels. Cross-section images were obtained by automatic layering of individual slices from each Z-stack. Red-Green-Blue (RGB) graphic profiles were created by analyzing the distribution and intensity of pixels of these colors along a chosen line using ImageJ software. All individual images were processed using Image J software and assembled for presentation using Photoshop and Illustrator software (Adobe).

## Calcium switch and transepithelial electrical resistance [TEER]

MDCK cells were plated at $1.5 \times 10^5$ cells per squared centimeter and allowed to establish confluent monolayers for 2–3 days before incubation for 16 hr in low-calcium medium (S-MEM containing 5% dialyzed FBS) and switched to normal-calcium medium (low-calcium medium supplemented with 1.8 mM CaCl$_2$) for the indicated times. TEER of MDCK cells grown on 12-mm polycarbonate Trans-well filters (0.4 µm pore size; Corning Costar, Corning, NY) were measured by using an epithelial vol-tohmmeter Millicel-ERS resistance meter (Millipore, Carlsbad, CA) with three parallel filters for each group of cells at each time point. TEER values were obtained by subtracting the blank values from the filters and the medium, and expressed in ohm·cm$^2$. For other experiments carried out in this work that required TEER measurements, various pharmacologic and chemical treatments were car-ried for the indicated durations (in each figure panel) prior to analysis using the same protocol out-lined above.

## MDCK cystogenesis assays

The 3D collagen matrix procedure was performed as described previously (*Bhandari et al., 2015*). Briefly, $3X\ 10^4$ MDCK cells were added to a collagen solution at pH 7 (GlutaMAX 24 mM, NaHCO$_3$ 2.35 mg/ml, DMEM 1X, FBS 2%, Hepes 20 mM, Collagen I 2 mg/ml) and placed in a well of a 4 well

chamber slide. After the collagen polymerization, culture media (DMEM 1X, FBS 2%) was added to the plate which was placed in a 37°C $CO_2$ incubator. Media was changed every 2 days for up to 2 weeks. Cysts and tubule formations were monitored by phase-contrast microscopy. *Cyst volumes were calculated* using the formula $4/3 \times \pi \times r^3$ assuming that *cysts* are spherical in shape.

### Anchorage-dependent colony growth assay

Anchorage-dependent growth was monitored on a solid (plastic) surface as described previously (*Lopez-Sanchez et al., 2015*). Briefly, approximately ~5000 DLD1 cells stably expressing various GIV constructs were plated in 6-well plates and incubated in a 5% $CO_2$ at 37°C for ~2 weeks in 0.2% FBS growth media. Colonies were then stained with 0.005% crystal violet for 1 hr. Each experiment was analyzed in triplicate.

### Anchorage-independent colony growth assay

Anchorage-independent growth of DLD1 cells was analyzed in agar as described previously (*Aznar et al., 2015*). Briefly, petri plates (60 mm) were pre-layered with 3 ml 1% Bacto agar (Life Technologies) in DMEM containing 10% FBS. Approximately ~5000 DLD1 cells stably expressing various GIV constructs were then plated on top in 3 ml of 0.3% agar–DMEM with 10% FBS. All assays were carried out using three replicate plates at a seeding density of ~5000 cells/plate. Following overnight incubation in 5% $CO_2$ incubator, 1 ml DMEM supplemented with 2% FBS was added to maintain hydration. After 2 weeks of growth, colonies were stained with 0.005% crystal violet/methanol for 1 hr, and subsequently photographed by light microscopy. The number of colonies in ~15–20 randomly-selected fields were counted at 10X magnification. Each experiment was analyzed in triplicate.

### Oncogenic Ras transformation assays

Neoplastic transformationin Ras-transformed NIH3T3 fibroblasts was analyzed using standard assays of colony formation in soft agar as described previously (*Clark et al., 1995*). Low passage NIH3T3 cells (~5000) stably co-transfected with appropriate GIV-Flag construct (2 μg cDNA) and HA-Ras G12V (1 μg cDNA) were analyzed for their ability to form tumor foci in soft agar plates. Plates were incubated in 5% $CO_2$ at 37°C for ~2 weeks in growth media supplemented with 2% FBS. They were finally incubated with 0.1% (wt/vol) 3-(4,5-dimethylthiazol-2-yl)2 2,5-diphenyl tetrazolium bromide (MTT; Sigma) in PBS for 1 hr to visualize colonies. The remaining NIH3T3 cells not used for this assay were lysed and analyzed for GIV-Flag and Ha-Ras G12V expression by immunoblotting.

### Confocal imaging and 3D reconstruction of deconvolved images

Confocal Laser scanning microscopy (CLSM) was performed with an Olympus FV3000 microscope equipped with a 60X oil immersion objective (Apochromat 60XOSC2, NA 1.4), with the pinhole set at 1 airy unit. Images were sequentially recorded with excitation wavelengths 405, 488 and 561 with the corresponding dichroic mirror. Z-stacks were acquired 0.2 mm spacing, according to Nyquist with a size of 1024 x 1024 pixels. Images were processed using the 3D deconvolution and 3D reconstructions tools of the Olympus cellSens (version Dimension Desktop 1.15) software. Co-localization analysis was performed using single stack or maximum projection images with the use of public domain Java image processing program ImageJ (version 1.49v, http//imagej.nih.gov/ij). Crop images were further sharpened in ImageJ.

### Tubulin co-sedimentation assays

The association of GIV with microtubules was monitored using a method adapted from (*Vallee, 1982*; *Vallee and Collins, 1986*). Briefly, microtubules were polymerized from whole-cell lysates of Cos7 cells transfected or not with WT-GIV, S245D-GIV or S245A-GIV. Cells were scraped off the tissue culture plates, and washed with chilled PBS. Of the cell pellet, 10% were used for Triton-X lysis to monitor total amounts of protein evaluated (inputs). The pellet was then washed again in chilled PEM buffer (0.1 M PIPES, 1 mM EGTA, 1 mM $MgSO_4$, pH 6.6) and lysed by Dounce homogenization in hypotonic buffer (1 mM EGTA, 1 mM $MgSO_4$, pH 6.6) supplemented with protease and phosphatase inhibitors, DTT and $Na_3VO_4$. After lysis, 0.1 M PIPES (pH 6.6) was added and lysates were centrifuged at 4°C for 10 min at 20,000X g. The supernatant was separated from the

cell pellet and was then clarified by centrifugation at 4°C for 60 min at 130,000X g. The supernatant was then incubated with 25 μM Docetaxel at 37°C for 30 min in the presence of 1 mM GTP to allow microtubules to polymerize. Microtubules were separated from soluble tubulin by centrifugation at room temperature for 30 min at 17,000X g. The supernatant, containing soluble tubulin, was collected and supplemented with 5X sample buffer. The pellet, which contained polymerized microtubules, was gently washed with PEM buffer and resuspended in 4X sample buffer in PEM. Protein in the supernatant (S), input (I) and pellet (P) were analyzed by immunoblotting.

## Interaction network analysis

Multiple databases of gene or protein interactions, such as those compiled by extensive PubMed search, iRefIndex (*Razick et al., 2008*), BioGRID (*Chatr-Aryamontri et al., 2015*), IntAct (*Kerrien et al., 2012*), STRING (*Szklarczyk et al., 2015*) or GeneMANIA (*Warde-Farley et al., 2010*) were mined for functional interactomes of the components of the energy-sending pathway (AMPK, LKB1 and the AMPK-related family of kinases) and GIV (ccdc88a). The genes and a subset of their neighbors were then extracted from the databases and reconstructed as an interaction network to highlight the major links between energy-sensing components, the junctional complex-associated genes, and the genes that provide input signals (growth factors, integrins and the Wnt pathway) for the assembly and disassembly of junctions. All splice isoforms, information regarding whether an interaction is stimulatory or inhibitory, or post-translational modifications are collapsed, i.e., each node represents all the proteins produced by a single, protein-coding gene locus, and each link represents some form of functional association.

## Statistical analysis

Each experiment presented in the figures is representative of at least three independent biological repeats (with at least two technical repeats for each condition within each biological repeat). Statistical significance between the differences of means was calculated by an unpaired student's t-test. A two-tailed pvalue of <0.05 at 95% confidence interval is considered statistically significant. All graphical data presented were prepared using GraphPad.

## Acknowledgements

We thank Gordon N Gill (UCSD) and Deepali Bhandari (CSULB) for their critical input during the preparation of the manuscript. This work was supported by NIH grants CA100768, CA160911 and DK099226 (to PG). PG. was also supported by the Burroughs Wellcome Fund (CAMS award), the American Cancer Society (ACS-IRG 70-002) and by the UC San Diego Moores Cancer Center. IK was supported by NIH grants GM071872 and AI118985. MG-F was supported by the NIH (CA100768). CCR was supported by the NCI/NIH (T32CA121938).

## Additional information

### Funding

| Funder | Grant reference number | Author |
|---|---|---|
| National Cancer Institute | Postdoctoral Fellowship T32CA121938 | Cristina C Rohena |
| National Institute of General Medical Sciences | R01GM071872 | Irina Kufareva |
| National Institute of Allergy and Infectious Diseases | R01AI118985 | Irina Kufareva |
| National Cancer Institute | R01CA100768 | Pradipta Ghosh Marilyn G Farquhar |
| National Cancer Institute | R01CA160911 | Pradipta Ghosh |
| National Institute of Diabetes and Digestive and Kidney Diseases | R01DK099226 | Pradipta Ghosh |

| American Cancer Society | ACS-IRG 70-002 | Pradipta Ghosh |

The funders had no role in study design, data collection and interpretation, or the decision to submit the work for publication.

## Author contributions

NA, Conception and design, Acquisition of data, Analysis and interpretation of data, Drafting or revising the article; AP, CCR, YD, LPJ, VT, IK, Acquisition of data, Analysis and interpretation of data; MGF, Contributed unpublished essential data or reagents; PG, Conception and design, Analysis and interpretation of data, Drafting or revising the article

## Author ORCIDs

Pradipta Ghosh, http://orcid.org/0000-0002-8917-3201

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
