## [Decision Letter]

Thank you for submitting your article "AMPK fortifies epithelial tight junctions during energetic stress via its effector GIV/Girdin" for consideration by *eLife*. Your article has been favorably evaluated by Randy Schekman (Senior Editor) and three reviewers, one of whom, Johanna Ivaska (Reviewer#1), is a member of our Board of Reviewing Editors. The following individuals involved in review of your submission have agreed to reveal their identity: Michael J Caplan (Reviewer #2); Tomi Mäkelä (Reviewer #3).

The reviewers have discussed the reviews with one another and the Reviewing Editor has drafted this decision to help you prepare a revised submission.

Summary:

This interesting and well written manuscript describes experiments designed to explore the role of GIV/Girdin in the stabilization of epithelial tight junctions. It has previously been shown that activation of the AMPK kinase both facilitates tight junction assembly and stabilizes tight junctions, enhancing their resistance to factors such as energy deprivation that ordinarily lead to tight junction dissolution. The authors provide evidence indicating that GIV/Girdin is a substrate for AMPK activity. Using antibodies directed against GIV/Girdin and phosphoGIV/Girdin they show that GIV/Girdin and phosphoGIV/Girdin are not present at tight junctions at steady state. Under conditions that produce energy stress or that destabilize junctions, however, phosphoGIV/Girdin accumulates at sites of cell-cell contact. They further report that overexpression of phosphomimetic GIV/Girdin maintains junction integrity and TEER in the face of juncetion-perturbing stimuli, whereas this effect is not seen with overexpression of a non-phosphorylatable GIV/Girdin. The authors suggest that AMPK-mediated phosphorylation of GIV/Girdin constitutes the mechanism through which AMPK mediates its effects on junction stability. They find that phosphoGIV/Girdin is associated in a phosphorylation-dependent manner with microtubules that are organized along cell contact sites, and suggest that this microtubule association may play a role in the effects of GIV/Girdin on junction stability. Some tumors express a mutant form of GIV/Girdin that lacks the AMPK phosphorylation site. The authors suggest that AMPK-mediated phosphorylation of GIV/Girdin may act to link the regulation of cell growth to cell junction stability and that expression of this mutant form of GIV/Girdin might allow tumor cells to grow in an anchorage-independent fashion. They find that expression of phosphomimetic GIV/Girdin prevents anchorage independent growth, whereas expression of non-phosphorylatable GIV/Girdin does not prevent anchorage-independent growth. The data are generally strong and well presented. The figures are clear and well presented. There are, however, several points that should be addressed and questions that need to be answered.

Essential revisions:

The data shown support the conclusion that Ser245 phosphorylation of GIV plays a role in TJ integrity. However, the current data does not sufficiently demonstrate that the observed TJ regulation by GIV is dependent on AMPK. Furthermore, almost all data relies on overexpression of GIV mutants. The report in whole would be significantly strengthened by providing evidence of endogenous GIV participating in energy stress signaling to tight junctions in an AMPK-dependent manner.

1) Absence and presence of energy stress and knock-down of AMPK should be shown to demonstrate the role of AMPK and energy stress in phosphorylating GIV S245 (Figure 1).

2) What is the localization of AMPK upon stress, does it translocate to the junctions?

3) Calcium deprivation / EGTA treatment should be linked to AMPK activity to demonstrate that AMPK-dependent phosphorylation is needed for proper cell contacts.

4) In Figure 8. The data should be shown of all mutants and the parental cell line in the presence and absence of energy stress (AMPK activation).

5) The imaging data presented in Figure 2, Figure 3, Figure 4 and Figure 5 provide critical support for the authors' interpretation of the role of GIV/Girdin. It is somewhat disappointing, therefore, that no effort is made to provide any quantitative analysis of these data. It would substantially enhance the impact of these data if the authors provided quantification and statistical analysis of the extent of localization of GIV/Girdin to sites of cell-cell contact under each of the conditions tested. Similarly, a quantitative analysis of the extent of co-localization of GIV/Girdin with occludin and β-catenin (as depicted in Figure 3) would be very valuable.

6) The lack of recovery of TEER in SA cells returned to NCM (Figure 2) is perplexing, since SA cells develop normal TEER at steady state. How do the authors explain this apparent discrepancy?

7) Similarly, why are SA cells (in which GIV is not available for AMPK phosphorylation) sensitive to Compound C? (5A). This is especially perplexing since the effects of Compound C on occludin localization (5D) are the same in SA and WT. The authors should address this apparent inconsistency. In addition, they should note and acknowledge that Compound C is not highly selective for AMPK and can have off target effects on a number of other kinases.

[Editors' note: further revisions were requested prior to acceptance, as described below.]

Thank you for resubmitting your work entitled "AMP-activated protein kinase fortifies epithelial tight junctions during energetic stress via its effector GIV/Girdin" for further consideration at *eLife*. Your revised article has been favorably evaluated by Randy Schekman (Senior Editor) and a Reviewing Editor Johanna Ivaska.

The manuscript has been improved but there are some remaining issues that need to be addressed before acceptance, as outlined below:

Data revisions or further clarifications needed for experimental data:

Reviewer point 4:

It seems that no revisions were made to this manuscript based on the comments and we apologize if the question was not clear. The request was to include the missing data from the parental cell line in 8H (no energy stress). If the glucose starvation is not feasible, using AICAR or metformin are a suitable alternative. However, the data with these treatments in 9A and B lack data with the parental cells and the GIV-SD mutant.

Reviewer point 6.

Please include some of this explanation for this discrepancy (as nicely explained in your response letter) in the manuscript text as you find appropriate.

Editorial revisions necessary:

Figure 1. Please include the AMPK wt and AMPK -/- labelling also on top of the immunoprecipitation panel (like you have done for the lower lysate panels). Please state in the figure legend how many experiments were performed with similar results (as you are only showing a representative blot and no quantifications).

Figure 2—figure supplement 2

Please state in the figure legend how many experiments were performed with similar results (as you are only showing a representative blot and no quantifications).

The word supplement is miss-typed "supplement" in several supplementary data figures, please correct.

---

## [Author Response]

[…]

*Essential revisions:*

*The data shown support the conclusion that Ser245 phosphorylation of GIV plays a role in TJ integrity. However, the current data does not sufficiently demonstrate that the observed TJ regulation by GIV is dependent on AMPK. Furthermore, almost all data relies on overexpression of GIV mutants. The report in whole would be significantly strengthened by providing evidence of endogenous GIV participating in energy stress signaling to tight junctions in an AMPK-dependent manner.*

This comment has two parts. In the first part, we are glad that the reviewers / editors are in agreement that "the data shown support the conclusion that Ser245 phosphorylation of GIV plays a role in TJ integrity". This is important to us because it is our major claim in this work, and much of our efforts went into comprehensive and conclusive experimentation to test this central hypothesis using multiple complementary approaches, generate phosphospecific Abs, use relevant mutants, etc.

In the second part, the reviewers/editors have pointed out that a major weakness is the need to overexpress GIV (WT or mutants) in our studies. They rightly point out that the study would be significantly strengthened by testing the role of endogenous GIV at TJs in an AMPK-dependent manner and go on to suggest a few experiments. We agree and have included in this revised submission the experiments suggested (see Responses #1 and #2 below) to implicate endogenous AMPK phosphorylates endogenous GIV under energetic stress, and that endogenous AMPK localizes to the TJs. As for our need to use of exogenously expressed GIV mutants in MDCK cell lines at levels equal amounts of WT endogenous GIV, we had addressed this technical limitation in detail in the subsection “Phosphorylation of GIV at S245 is essential for junctional integrity and epithelial morphogenesis” explaining why this was done, how we validated the cell lines to confirm that such expression did not affect the functional and structural integrity of junctions at steady-state.

*1) Absence and presence of energy stress and knock-down of AMPK should be shown to demonstrate the role of AMPK and energy stress in phosphorylating GIV S245 (Figure 1).*

We have now carried out the suggested experiment and included the results in Figure 1 as panel I. Consistent with our prior findings, we confirmed that endogenous AMPK is necessary and sufficient to trigger phosphorylation of endogenous GIV at S245. More specifically, we found that energetic stress (induced by glucose depravation) could trigger phosphorylation of endogenous GIV at S245 in mouse embryonic fibroblasts (MEF) from control mice, but not in MEFs from AMPK knockout littermates (AMPK -/-) [see subsection “AMPK binds and phosphorylates GIV at residue S245”, second paragraph, and Figure 1 legend].

*2) What is the localization of AMPK upon stress, does it translocate to the junctions?*

We have now carried out the suggested experiment. We analyzed total-AMPK and phospho(active)-AMPK in domed monolayers of MDCK cells as they are subjected to energetic stress by glucose deprivation. These conditions mirror the exact conditions showcased in Figure 2 [in which we had analyzed the distribution of pS245-GIV]. We found that (phospho)-AMPK was not detected in fully polarized domed monolayers at steady-state grown in the presence of glucose (no energetic stress); however, when domed monolayers are subjected to 6 h of energetic stress [induced by glucose starvation] active AMPK was detected exclusively at the TJs of tricellular contact (tTJs), as determined by colocalization with Occludin where 3 or more cells come in contact. With prolonged energetic stress (i.e., 12 h of glucose starvation) tTJs were the first to disassemble and were associated with a blush of active AMPK. Longer duration of glucose starvation was associated with a loss of any specific signal for active AMPK. In this revised submission, we have now included our findings of localization of phospho-AMPK as Figure 2—figure supplement 1 (see subsection “GIV phosphorylated at S245 localizes to cell-cell junctions” and Figure 2—figure supplement 1 legend).

Unfortunately, we could not detect endogenous total AMPK under any of these conditions despite trying various commercially available antibodies. This is perhaps not unexpected because other groups who discovered the role of AMPK as a master regulator of the 'stress polarity pathway' in 2006-2007 [both Cantley and Caplan groups] also did not report the localization of AMPK at junctions.

We thank the reviewers/editors for this question because the experiments we have carried out have indeed revealed some unexpected new findings in the context of AMPK and raise the possibility that specific cellular components in the found exclusively at the tricellular TJs may either activate AMPK at that location or help localize activated AMPK to that site.

*3) Calcium deprivation / EGTA treatment should be linked to AMPK activity to demonstrate that AMPK-dependent phosphorylation is needed for proper cell contacts.*

We agree. In this revised submission we added an additional Figure [Figure 2—figure supplement 2] in which we have displayed the immunoblots corresponding to the immunofluorescence studies in Figure 2. These immunoblots confirm that phosphorlation of GIV at S245 after calcium deprivation (in the presence of EGTA) or during energetic stress (induced by glucose deprivation) is accompanied by phosphorylation and activation of AMPK. See subsection “GIV phosphorylated at S245 localizes to cell-cell junctions” and Figure 2—figure supplement 2 legend.

*4) In Figure 8. The data should be shown of all mutants and the parental cell line in the presence and absence of energy stress (AMPK activation).*

In this comment, the reviewer refers to our data [anchorage-dependent and independent growth] on DLD1 colorectal cancer cells. We had compared and contrasted the two modes of growth [in 3D vs. 2D] precisely because one, i.e., 3D growth, but not 2D growth assay is known to recreate in culture the typical 3D architecture of the tumor tissue and heterogeneously exposes cancer cells to diffusion-limited glucose, oxygen and nutrients and to other physical and chemical stresses at the tumor core during the early periods of avascular growth. When we induced further energetic stress [by glucose starvation] of DLD1 cells in 3D growth, this led to complete growth arrest of all GIV-expressing stable cell lines tested, making it impossible to assess differences. This is not unexpected given the fact that these tumor cells, like all others, are chronically energy [ATP] depleted despite ~10-15 fold higher glucose consumption [i.e., Warburg effect]. Therefore, it is not surprising that glucose deprivation was incompatible with growth. We would like to point out that we have indeed carried out the experiment that is proposed here, but using a slightly different approach. Instead of glucose deprivation as a way to activate AMPK in these anchorage-independent growth assays, we activated AMPK with AICAR or Metformin, and those results were showcased in Figure 9 (panels 9A and 9B). It is certainly possible that we have misinterpreted the question, and if so, we are willing to explain/respond with further feasible experimentation.

*5) The imaging data presented in Figure 2, Figure 3, Figure 4 and Figure 5 provide critical support for the authors' interpretation of the role of GIV/Girdin. It is somewhat disappointing, therefore, that no effort is made to provide any quantitative analysis of these data. It would substantially enhance the impact of these data if the authors provided quantification and statistical analysis of the extent of localization of GIV/Girdin to sites of cell-cell contact under each of the conditions tested. Similarly, a quantitative analysis of the extent of co-localization of GIV/Girdin with occludin and β-catenin (as depicted in Figure 3) would be very valuable.*

In this comment, the reviewer asked us to provide a quantification of some sort for the IF panels showcased in Figure 2, Figure 3, Figure 4 and Figure 5. As an example, he/she asked us to assess the extent of colocalization using RGB profiler as we have done in Figure 3. We agree. In Figure 2 and Figure 3 of the original submission, we had presented quantification of colocalization using graphed intensities of 'red' and 'green' pixels using the RGB profile graphical analysis tool on ImageJ. We have now provided the quantitative analysis of the remaining panels listed in this comment (Figure 4 and Figure 5) and included the additional data in Figure legends for these two panels indicating what% of surface area of how many random fields imaged had the phenotype that is displayed in the Figure panels.

As for the quantification of the extent of colocalization of total GIV/Girdin in the Figure 2, Figure 3, Figure 4 and Figure 5, we could not do that reliably and consistently because the currently available anti-GIV antibodies, although able to detect a small pool of GIV at junctions after energetic stress, the signal is faint and seen only transiently amidst a larger pool that is known to be mostly cytosolic [see Figure 2]; such noisy faint signals made quantification of colocalization unreliable and almost impossible. This was the primary reason we invested in generating and validating a phosphospecific GIV antibody that can specifically detect the specialized pool of GIV at the TJs. We have confirmed that both SA and SD mutant GIV cannot be detected with anti-pS245GIV antibody [see Figure 1], and hence, we did not use this antibody in the IF assays characterizing the role of phosphorylation of GIV at TJs using the various cell lines [Figure 4, Figure 5]; instead we just followed the disassembly of junctions using occludin and E-cadherin as markers of TJs and AJs, respectively.

*6) The lack of recovery of TEER in SA cells returned to NCM (Figure 2) is perplexing, since SA cells develop normal TEER at steady state. How do the authors explain this apparent discrepancy?*

In this question, the reviewer accurately points out that GIV-SA expressing cells have normal TEER at steady-state [no energetic stress, normal calcium, normal serum], but noted that they did not 'recover' TEER when they were returned to normal-calcium growth conditions after an initial low-calcium exposure. We too have been perplexed by this observation. We believe that the prolonged delay in [almost lack of] recovery could be because the cells expressing the non-phosphorylatable GIV-SA mutant is expected to be defective in two conditions where we showed that the AMPK-GIV pathway is triggered: 1) when single cells come in contact and assemble TJs [see Figure 2, top two panels]; and 2) specifically at TJs subjected to Calcium or glucose deprivation [see Figure 2, lower three panels]. So, after the junctions are disassembled in GIV-SA cells [drop in TEER in low-calcium state] single cells must come together in normal calcium medium. However, because these cells are unable to phosphorylate GIV at S245 as single cells come together during the reformation of the monolayer, we believe that some of these cells just detach and die during the prolonged assay period [as seen in IF assays (Figure 5) and by light microscopy (Figure 5—figure supplement 1]. We speculate that energetic stress or disassembly of junctions due to low calcium may set in motion some yet unknown pathways in an irreversible manner after junctions are disassembled in the absence of the AMPK-GIV axis [in GIV-SA cells].

*7) Similarly, why are SA cells (in which GIV is not available for AMPK phosphorylation) sensitive to Compound C? (5A). This is especially perplexing since the effects of Compound C on occludin localization (5D) are the same in SA and WT. The authors should address this apparent inconsistency. In addition, they should note and acknowledge that Compound C is not highly selective for AMPK and can have off target effects on a number of other kinases.*

In this comment the reviewer points to the findings in 5A where we show that the functional integrity of TJs in GIV-SA cells is more significantly impacted after treatment of cells with Compound C than GIV-WT cells [i.e., drop in TEER is more significant]. He/she also rightly points out that in panel 5Dthe effect of Compound C on the structural integrity of TJs and AJs [as determined by IF] looks similar, i.e., in both cases junctional proteins are lost from cell-cell contact sites by 6h after treatment with Compound C. We would like to point out that TEER measurements had to be carried out with shorter periods of incubation [2 h], whereas IF assays were carried out with longer incubation [3, 6, 10 and 20 h]. Shorter incubations were necessary to see the early impairment in functional integrity [TEER] and tease out the differential sensitivity of WT and SA cells. Longer incubations in IF assays that were designed to look at structural integrity of TJs/AJs were more helpful to appreciate the resistance of GIV-SD cells, but were not sensitive enough to see early losses in functional integrity. In order to make sure that readers notice the difference of incubation times [which we mention in the labels for each Figure panel], we have now reinforced the short and prolonged exposures in the subsection “Stabilization of tight junctions by AMPK requires phosphorylation of GIV at S245”, just so that readers understand that the data is consistent overall.

As for the fact that Compound C may have off target effects beyond just inhibition of AMPK, this is a very important point that we have now included in the subsection “Stabilization of tight junctions by AMPK requires phosphorylation of GIV at S245” when this compound was used for the first time. We thank the reviewer for pointing this out.

[Editors' note: further revisions were requested prior to acceptance, as described below.]

[…]

*The manuscript has been improved but there are some remaining issues that need to be addressed before acceptance, as outlined below:*

*Data revisions or further clarifications needed for experimental data:*

*Reviewer point 4:*

*It seems that no revisions were made to this manuscript based on the comments and we apologize if the question was not clear. The request was to include the missing data from the parental cell line in 8H (no energy stress). If the glucose starvation is not feasible, using AICAR or metformin are a suitable alternative. However, the data with these treatments in 9A and B lack data with the parental cells and the GIV-SD mutant.*

We had misunderstood this question in the first revision cycle. We appreciate this clarification. We have now included the data with DLD1 parental cells in panels 9Aand 9B. These cells, much like the DLD1 cells expressing GIV-WT, are susceptible to the tumor-suppressive actions of Metformin and AICAR. The text and Figure legend are also edited accordingly. The DLD1-GIV-SD cells virtually do not form colonies in 3D anchorage-independent growth assays [please see Figure 8]; treatment with AICAR/Metformin did not change that finding. We left out DLD1-GIV-SD from the graphs displayed in 9A-Bto avoid confusing readers because display of the SD data [normalized to untreated, as done for the remaining cell lines] will show that colony growth [~0-4 colonies/well] remain at 100% despite treatment with AMPK agonists.

Reviewer point 6.

*Please include some of this explanation for this discrepancy (as nicely explained in your response letter) in the manuscript text as you find appropriate.*

We have now included the stated explanation in the subsection “GIV phosphorylated at S245 localizes to cell-cell junctions”.

*Editorial revisions necessary:*

*Figure 1. Please include the AMPK wt and AMPK -/- labelling also on top of the immunoprecipitation panel (like you have done for the lower lysate panels). Please state in the figure legend how many experiments were performed with similar results (as you are only showing a representative blot and no quantifications).*

We have now revised Figure 1 and legend to do exactly as recommended.

Figure 2—figure supplement 2

*Please state in the figure legend how many experiments were performed with similar results (as you are only showing a representative blot and no quantifications).*

We have now revised the legend for Figure 2—figure supplement 2 to indicate how many times we have assessed AMPK phosphorylation when EGTA or glucose starvation were used as ways to activate AMPK in various assays.

*The word supplement is miss-typed "supplement" in several supplementary data figures, please correct.*

We apologize for this error. We have now revised 2 supplementary figures where the word supplement was misspelled.